

# Thermal regime of the Grigoriev ice cap and the Sary-Tor glacier in the Inner Tien Shan, Kyrgyzstan

Lander VAN TRICHT[1], Philippe HUYBRECHTS[1]

[1]Earth System Science and Departement Geografie, Vrije Universiteit Brussel, Brussels, Belgium

*Correspondence to*: Lander Van Tricht (lander.van.tricht@vub.be)

**Abstract.** The thermal regime of glaciers and ice caps represents the internal distribution of ice temperatures. Accurate knowledge of the thermal regime is important to understand the dynamics and response of ice masses to climate change, and
to model their evolution. Although the assumption is that most ice masses in the Tien Shan are polythermal, this has not been examined in appropriate detail so far. In this research, we investigate the thermal regime of the Grigoriev ice cap and the Sary-Tor glacier, both located in the Inner Tien Shan in Kyrgyzstan, using a 3D higher-order thermomechanical ice flow model. Input data and boundary conditions are inferred from a surface energy mass balance model, a historical air temperature and precipitation series, ice thickness measurements and reconstructions, and digital elevation models. Calibration and validation
of the englacial temperatures is performed using historical borehole measurements on the Grigoriev ice cap and radar measurements for the Sary-Tor glacier. The results of this study reveal a polythermal structure of the Sary-Tor glacier and a cold structure of the Grigoriev ice cap. The difference is related to the larger amount of snow (insulation) and superimposed ice (release of latent heat) for the Sary-Tor glacier resulting in higher ice surface temperatures, especially in the accumulation area, which are subsequently advected downstream. Further, ice velocities are much lower for the Grigoriev ice cap with
consequent lower horizontal advection rates. A detailed analysis concerning the influence of temperature and precipitation changes at the surface reveals that the thermal structure of both ice bodies is not a constant over time, with recent climate change causing more temperate ice in higher areas. The selected ice masses are representative examples of the (Inner) Tien Shan glaciers and ice caps. Therefore, our findings and the calibrated parameters can be generalised allowing to improve the understanding of the dynamics and future evolution of other glaciers and ice caps in the region.

## 1 Introduction

Global warming is causing an unprecedented loss of glaciers that is expected to continue and have far-reaching consequences for worldwide water availability, even without further temperature increase (Zemp et al., 2019). High Mountain Asia (HMA), comprising several glacierised mountain ranges such as the Himalayas, Karakoram, Pamir and Tien Shan, contains the highest concentration of glaciers globally (Bhattacharya et al., 2021). However, glaciers in HMA also involve one of the largest
uncertainties in current and future ice volume assessments (Farinotti et al., 2019). An important characteristic influencing the





dynamics and behaviour of glacier ice is the englacial temperature (Gilbert et al., 2020). Accurate knowledge of the thermal regime of ice bodies is therefore needed to understand the response of ice bodies to climate change.

The focus of this study is on the Tien Shan, where ice masses are assumed to be mostly cold and polythermal (Vilesov, 1961; Maohuan et al., 1982; Echelmeyer and Zhongxiang, 1987; Vasilenko et al., 1988; Petrakov et al., 2014; Nosenko et al., 2016). The latter means that the ice mass consists of both warm/temperate ice, i.e. the ice temperature reaches the pressure melting point, and cold ice. Previous studies demonstrated a polythermal structure using borehole measurements for the Ürümqi glacier (Cai et al., 1988) and the Davydov glacier (Vasilenko et al., 1988). Using GPR measurements, other studies revealed the presence of cold and warm layers on the Abramov glacier (Kronenberg et al., 2020), the Sary-Tor glacier (Petrakov et al.,

2014) and the Tuyuksu glacier (Nosenko et al., 2016). According to measurements on the Abramov glacier, the Ürümqi glacier and the Tuyuksu glacier, temperate ice was present near the surface of the lower part of the accumulation area (Kronenberg et al., 2020), and over a large area at the base (Nosenko et al., 2016). The latter was corroborated by larger summer velocities compared to winter velocities indicating that basal sliding was occurring on the Ürümqi glacier in the Chinese Tien Shan (Maohuan et al., 1989,1992). Part of the Inner Tien Shan glaciers have also been categorised as surge-type glaciers (Mukherjee

et al., 2018; Bhattacharya et al., 2021). The surge behavior has been interpreted by the polythermal structure of the ice masses with a cold mantle at the front restraining ice movement. After a long period with ice thickening and steepening in the upper reaches, the cold mantle at the front is supposed to be destructed, followed by an advance over a short period of time (Nosenko et al., 2016; Mukherjee et al., 2017). Few (11) of such glaciers were identified in the Ak-Shyirak massive, including the Northern Bordu glacier and the Davydov glacier, located in the valleys next to the Sary-Tor glacier (Bondarev, 1961;

Mukherjee et al., 2017).

The temperate ice in the lowest part of the accumulation area has typically been explained by an infiltration recongelation zone, characteristic for the continental climate in which the ice masses are located (Vilesov, 1961; Maohuan et al., 1982,1990; Osmonov et al., 2013; Petrakov et al., 2014; Kronenberg et al., 2020). In the infiltration zone, which is the lowest part of the

accumulation area, the ice and snow surface is warmed significantly by the infiltration and refreezing of thawed meltwater releasing latent heat which can heat up the layers to zero-degree (Lüthi et al., 2015). Such a warm area in the infiltration zone has also been observed at other sub-polar glaciers such as Storglaciären in Sweden (Hooke et al., 1983). But, in addition to the influence of refreezing meltwater, other processes play a role in the warming and cooling of surface layers, such as a seasonal snow cover that shields the underlying ice from the extremely cold air temperatures during winter (Hooke et al., 1983) or the

ablation of ice which removes the warm or cold ice from above (Blatter et al., 1987; Wohlleben et al., 2009).

Here, we investigate the thermal regime of the Grigoriev ice cap and the Sary-Tor glacier, both located in the Inner Tien Shan, Kyrgyzstan (Central-Asia), using a 3-dimensional higher-order thermomechanical ice flow model coupled to a surface energy mass balance model. For the Grigoriev ice cap, ice thickness measurements were performed in August 2021 with a ground





penetrating radar (GPR) system and used for a glacier wide ice thickness reconstruction. Ice temperatures derived from ice cores made in 1962 (Dikikh, 1962), 1990 (Thompson et al., 1993), 2001 (Arkhipov et al., 2004), and 2007 (Takeuchi et al., 2014), distributed over the ice cap, are used as calibration and validation for the thermal structure. Concerning the Sary-Tor glacier, observed and reconstructed ice thicknesses from 2013 are adopted (Petrakov et al., 2014), and radargrams displaying cold and temperate ice are used for the validation of the thermal structure. Different experiments are performed under constant

1960-1990 average climatic conditions. Afterwards, multiple sensitivity experiments are carried out to determine the influence of parameter choice, to assess the uncertainty of the results, and to determine the impact of changing climatic conditions at the surface on the thermal structure of both ice bodies.

**2 Study area and data**

**2.1 The Grigoriev ice cap and the Sary-Tor glacier**

The Grigoriev ice cap is a small ice cap (also called flat-top glacier) located at the southern slopes of the Terskey Ala-Too mountain range in the Inner Tien Shan in Kyrgyzstan (Central Asia) (Fig. 1). The ice cap is located between the valleys of the Chontor glacier (to the west) and the Popova glacier (to the east). The top of the Grigoriev ice cap is a flat snow field at an elevation between 4500 and 4600 m a.s.l. In contrast to a typical valley glacier, the ice of the Grigoriev ice cap flows in multiple directions, ending at steeper glacier fronts and at multiple smaller outlet glaciers (Fig. 1). The northern boundary of the

Grigoriev ice cap is also characterised by ice cliffs where mass is lost due to calving. The Grigoriev ice cap has been subject to numerous glaciological investigations such as mass balance measurements (Mikhalenko, 1989; Fuijita et al., 2011) and ice core drilling (Dikikh, 1965; Thompson et al., 1993; Arkhipov et al., 2004; Takeuchi et al., 2014; Takeuchi et al., 2019). In August 2021, we measured the ice thickness of the Grigoriev ice cap with a GPR system, following the approach of Van Tricht et al. (2021a).


The Sary-Tor glacier on the other hand is a small valley glacier, located in the north-western part of the Ak-Shyirak massif, roughly 30 km to the southeast of the Grigoriev ice cap (Fig. 1). Its ice thickness was measured in 2013 with a GPR system showing a maximum ice thickness of almost 160 m (Petrakov et al., 2014). Annual mass balance measurements on this glacier are performed since 2014 and these are deposited in the World Glacier Monitoring Surface (WGMS) database. Because debris

cover is almost entirely absent on the Sary-Tor glacier, the glacier changes are assumed to directly reflect the changing climate in the Inner Tien Shan (Petrakov et al., 2014). The Sary-Tor glacier has therefore typically been used as a reference glacier for the surrounding area, especially for the Ak-Shyirak massive (Mikhalenko, 1993). Van Tricht et al. (2021b) calibrated a surface energy mass balance model for the Sary-Tor glacier and reconstructed the historical mass balance between 1750 and 2020, showing a significant decrease in the mean specific mass balance, especially since the seventies. Recently, the concession of

the Kumtor gold mine, which is located north of the glacier (Fig. 1), was extended towards the valley of the Sary-Tor glacier, removing the terminal moraines of the Little Ice Age (LIA).





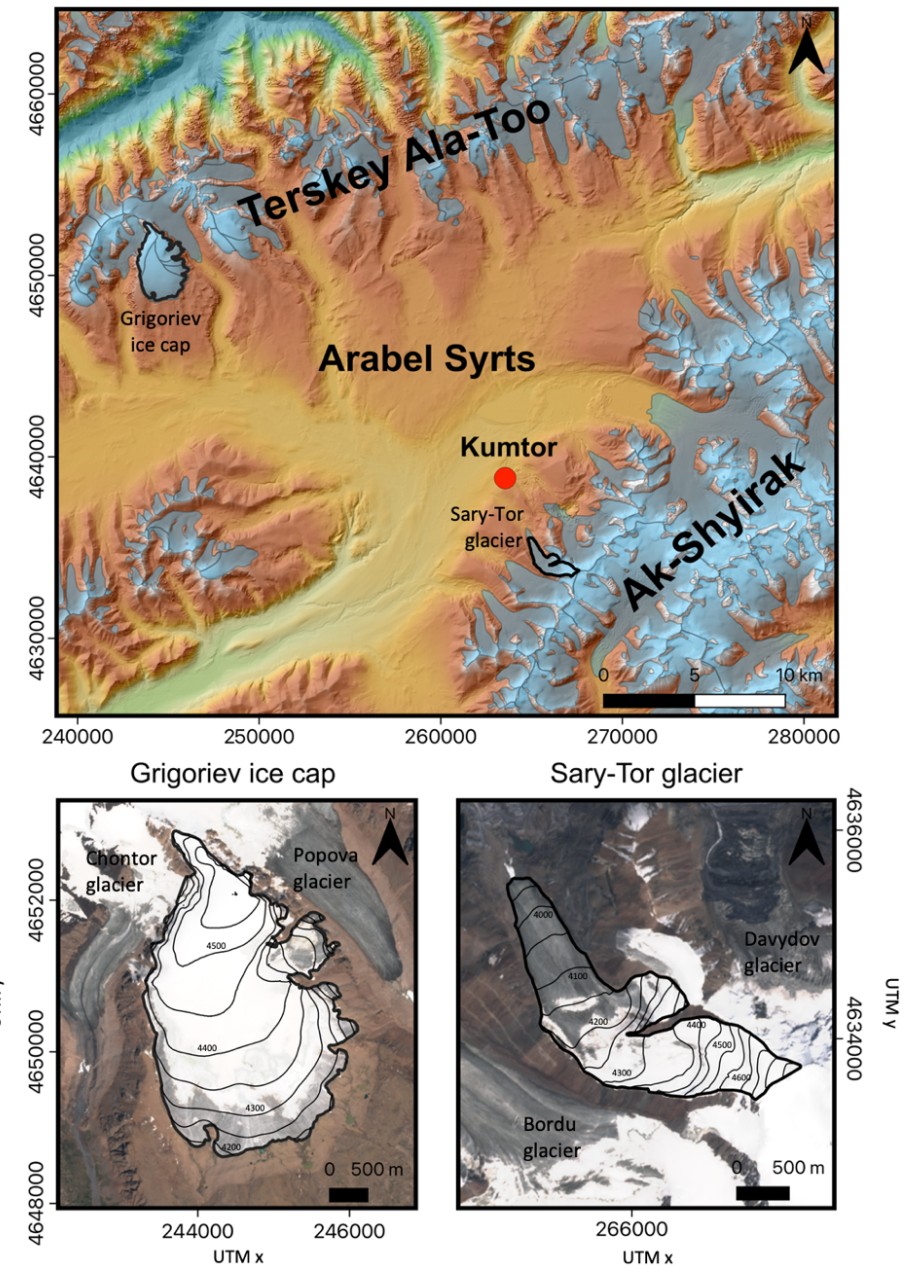

**Figure 1: The Grigoriev ice cap and the Sary-Tor glacier located in their respective mountain range in the Inner Tien Shan. All the glaciers of the Randolph Glacier Inventory version 6 (from 2002) are indicated in a light blue colour. The black outline of the glaciers represents the extent of the ice mass in August 2021. Glacier elevation contours are added for every 50 metres. The location of the Kumtor-Tien Shan meteorological station is added with a red dot. The background topography is from TanDEM-X, the satellite images are from Sentinel-2 on the 26th of July 2021.**



## 2.2 Englacial temperature measurements

Englacial temperatures represent the internal distribution of the ice temperatures of ice bodies. The thermal regime is universally categorised into three types: cold (containing cold ice, possibly with a thin layer of temperate ice over the base), temperate (containing only temperate ice), and polythermal (containing cold and temperate ice) (Blatter and Hutter, 1991). Accurate knowledge of the ice temperatures is important because ice deformation depends on the ice temperature, meltwater routing is affected by ice temperature and the basal temperature determines whether the ice mass can slide over its base. Englacial temperatures can be measured directly using boreholes. However, the distinction between cold and temperate ice can also be recognised on radargrams (Delf et al., 2022). Over the past 60 years, various temperature measurements have been carried out in the study area.

In 1962, a first drilling campaign was carried out on the Grigoriev ice cap between 4170 and 4420 m a.s.l. (Dikikh, 1965). The measurements revealed the presence of cold ice. In July 1985, the USSR Academy of Sciences performed a thermal drilling in the central part of the Davydov glacier, located in the valley next to the Sary-Tor glacier, at about 3950 m a.s.l. up to a depth of 102 m (Vasilenko et al., 1988). Temperatures were measured up to a depth of 30 m. The results showed that ice temperatures decreased sharply to almost -6°C at a depth of 6 m. Further down, ice temperatures increased quickly, reaching -0.2°C at a depth of 30 m. As such, the results strongly suggested a two-layer structure with a warm ice core over the bed. In 1990, a new drilling of a shallow ice core (16.5 and 20 m) was carried out at the summit of the Grigoriev ice cap, covering the last 50 years. (Thompson et al., 1993). Thompson et al. (1993) measured an ice temperature of -2°C at a depth of 20 m, substantially higher than the 1962 measured temperatures (Dikikh, 1965). One of the conclusions of the study of Thompson et al (1993) was that the 10 to 20 m temperatures generally exceeded the mean annual air temperature because of the refreezing of meltwater, and that ice temperatures increased strongly between 1960 and 1990. Thompson et al (1993) also observed a significant amount of melting and runoff during the 1990 summer season, as well as a few large pools of standing water on flat areas of the ice cap below 4500 m elevation. Such supraglacial water suggested cold, impermeable ice.

In September 2001, Arkhipov et al. (2004) made a drilling of 21.5 meter on the summit of the Grigoriev ice cap. During their survey, they noticed the presence of a thick water saturated firn layer by the end of the ablation period leading to a significant warming of the upper layers. Besides, Arkhipov et al. (2004) measured an abrupt ice temperature increase of several degrees over the transition from the ablation region to the accumulation region (at the EL, around 4300 m), which they explained by the heat released from refreezing melt water.

Finally, in September 2007, a last drilling was performed close to the top of the ice cap (4563 m a.s.l.) up to the bedrock (87.46 m) (Takeuchi et al., 2014). Temperature measurements of the drilled ice revealed that the ice was −2.7°C at 10 m deth and



down to $-3.9\ °C$ at the bottom. Hence, the basal ice was not at the pressure melting point (Takeuchi et al., 2014). The lower temperatures at the bottom of the ice suggested a recent warming of the top layers that had not yet reached the bottom.

On the Sary-Tor glacier, no direct ice temperature measurements were made. However, the radar measurements of Petrakov et al. (2014) clearly showed a distinction between warm ice over the bed and cold ice above at a surface elevation of 4150 m. This demonstrated a polythermal structure of this glacier.

## 3 Methods

### 3.1 Higher-Order glacier model

A 3-dimensional (3D) glacier model resolves the spatially distributed [x,y,z]-velocity fields in an ice mass. In this study, we apply a Blatter-Pattyn type of a 3D higher-order (HO) thermomechanical ice flow model to derive the horizontal velocities (Fürst et al., 2011). The HO model includes longitudinal and transverse stress gradients, which are neglected in the SIA approximation (Zekollari et al., 2013). Compared to a Full Stokes solution, the HO model assumes cryostatic equilibrium in the vertical, neglecting the vertical resistive stresses (bridging effects).


The selected ice flow model has been successfully applied at different scales, ranging from small mountain glaciers (Zekollari et al., 2013; Zekollari and Huybrechts, 2015), a medium-sized ice cap (Zekollari et al., 2017) to entire ice sheets (Fürst et al., 2013, 2015). The problem of determining velocities in 3D is reduced to determining only the horizontal velocities, because the vertical velocities are obtained through incompressibility. Nye's generalisation of Glen's flow law, which links deviatoric

stresses $(\tau_{ij})$ to strain rates $(\dot{\varepsilon}_{ij})$ (deformation and velocity gradients) (Eq. 1), is used to quantify the deformation (and the corresponding flow velocity due to internal deformation) of the ice based on the stresses that act on it.

$$\tau_{ij} = 2\eta\dot{\varepsilon}_{ij} \tag{1}$$

$$\eta = \frac{1}{2}A(T)^{-\frac{1}{n}}(\dot{\varepsilon}_e + \dot{\varepsilon}_0)^{\frac{1}{n}-1} \tag{2}$$

$$\dot{\varepsilon}^2{}_e = \frac{1}{2}\dot{\varepsilon}_{ij}\dot{\varepsilon}_{ij} \tag{3}$$

$$\dot{\varepsilon}_{ij} = \frac{1}{2}\left(\frac{\partial_i}{u_j} + \frac{\partial_j}{u_i}\right) \tag{4}$$

The symbols 'i' and 'j' denote the space derivative with respect to the $i^{th}$ and $j^{th}$ spatial components. n is the power-law exponent which is set to 3. $\eta$ is the effective viscosity of the ice (resistance to deformation) (2), which is defined via the second

invariant of the strain rate tensor (3). $\dot{\varepsilon}_0$ is a small offset $(10^{-30})$ that ensures finite viscosity (Fürst et al., 2011). The flow rate factor A(T) is a function of temperature (see sect. 3.2). The strain rate tensor is defined in terms of velocity gradients (4) with





u the 3D components of the velocity vector. We use a grid with a spatial resolution of 25 m, and 21 layers in the vertical are considered. The 3D HO model has a temporal resolution of approximately one week (0.02 year).

In the model, a Weertman-type sliding law is implemented in which the basal sliding velocity ($u_b$) is proportional to the basal drag ($\tau_b$) to the third power. This is a common approach in glacier modelling (Jouvet et al., 2011; Zekollari et al., 2013).

$$u_b = -A_{sl}\tau_b^3 \tag{5}$$

The basal drag is calculated following the HO approximation and corresponds to the sum of all basal resistive stresses. $A_{sl}$ is the sliding parameter which is defined as a constant in the model. When the basal temperature is below the pressure melting point, the ice is frozen to the bedrock and there is no basal sliding.

Knowing all the velocities in 3D (u), the continuity equation (Eq. 6) is used to link the ice dynamics and the surface mass
balance ($m_s$) (see sect. 3.4) to calculate the evolution of the ice thickness (H) over time (t). $\vec{u}$ is the vertically averaged horizontal velocity vector.

$$\frac{\partial H}{\partial t} = -\nabla(\vec{u}H) + m_s \tag{6}$$

### 3.2 3D temperature calculation and influence on flow rate factor

A full 3D calculation of the ice temperatures is performed simultaneously with the velocity calculations. This is crucial because ice temperature regulates the ice stiffness (viscosity) (see Eq. 2 and Eq. 6) and determines whether basal sliding occurs. The flow rate factor, A(T), affects the speed at which ice deforms under a given stress. Higher values comprise lower viscosities and hence faster deformation. Hence, the deformation rate increases with increasing temperatures because dislocations in the ice become more mobile. The temperature dependence is described using an Arrhenius relationship. At each time step, the ice
temperature is used to obtain the local flow rate factor using (Eq. 7).

$$A(T) = m\, a \exp(-\frac{Q}{RT_{pmp}}) \tag{7}$$

$T_{pmp}$ is the ice temperature corrected for pressure melting (Eq. 8), which is also called the homologous temperature.

$$T_{pmp} = T - \gamma H \tag{8}$$



With $\gamma$ being $8.7\times10^{-4}$, H the ice thickness and T the ice temperature. Hence, for increasing pressure, the ice potentially starts melting at sub-zero temperatures. As such, the ice's melting point decreases with increasing pressure. E.g. ice starts to melt at

-0.1°C below an ice column of 115 metres. For temperatures lower than -10°C, a = $1.14\times10^{-5}$ Pa$^{-n}$ a$^{-1}$ and Q = 60 kJ mol$^{-1}$ while for temperatures larger than -10°C, a = $5.47\times10^{10}$ Pa$^{-n}$ a$^{-1}$ and Q = 139 kJ mol$^{-1}$. m is an enhancement factor which is typically used for tuning. The tuning of the enhancement factor is an implicit way to include the softening effect due to factors such as water content and impurity content (Huybrechts et al., 1991). Q is the activation energy for creep and R is the gas constant (8.31 J mol$^{-1}$ K$^{-1}$).


The ice temperature is determined from vertical diffusion, three-dimensional advection and the heating due to deformation and basal sliding (Eq. 9). Diffusion depends on the specific heat capacity and the thermal conductivity of the ice, which depends on local ice temperature. Advection (horizontal and vertical) depends on the 3D velocity fields ($\vec{V}$ is the 3D velocity vector). Internal heat production (P) is caused by internal deformation and basal sliding.


$$\frac{\partial T}{\partial t} = \frac{k}{\rho c_p} \nabla^2 T - \vec{V}.\nabla T + P \tag{9}$$

k is the thermal conductivity; $c_p$ is the specific heat capacity. They both are functions of the temperature (Eq. 10 and Eq. 11). $\rho$ is the ice density (910 kg m$^{-3}$).


$$c_p = 146.3 + 7.253\, T(K) \qquad \text{[in J kg}^{-1}\text{ K}^{-1}] \tag{10}$$
$$k = 9.828 * e^{-0.0057 T(K)} \qquad \text{[in W m}^{-1}\text{ K}^{-1}] \tag{11}$$

The vertical velocity, needed for the vertical advection, is obtained through vertical integration of the incompressibility

condition from the base of the g lacier to a height z (Eq. 12), using the kinematic boundary conditions at the bed (Eq. 13). $u_b$ and $v_b$ are the horizontal velocity components at the bedrock, which correspond to the basal sliding components.

$$w_z = w_{z=h-H} - \int_{z=h-H}^{z=h} \left( \frac{\partial u}{\partial x} + \frac{\partial v}{\partial y} \right) dz \tag{12}$$
$$w_{z=h-H} = u_b \frac{\partial(h-H)}{\partial x} + v_b \frac{\partial(h-H)}{\partial y} - m_b \tag{13}$$


The bedrock elevation is assumed to be constant in time. h is the surface elevation (m), $m_b$ is the basal melt rate, which is positive in case of melting.



### 3.3 Boundary conditions

The calculation of the ice temperature distribution requires the specification of two boundary conditions: the ice temperature of the surface layer and the temperature gradient at the bottom of the ice mass.

### 3.3.1 Basal boundary conditions

At the bottom, heat is produced by geothermal heating (energy received by the ice at its base) and friction induced by basal sliding and ice deformation. Heat flow studies at regional scales in the Kyrgyz Tien Shan indicate the presence of a "cold

Cenozoic orogeny" with no additional heating associated to tectonic deformation and basin development. Therefore, the area is characterised by a relatively low geothermal heat flux of approximately 50 mW/m$^2$ (Bagdassarov et al., 2011; Delvaux et al., 2013; Barandun et al., 2015). This is in the same order as the value used for the modelling of glacier No.8 in the Hei valley in China (Wu et al., 2019) and for Abramov glacier (Barandun et al., 2015). This corresponds to a temperature gradient of 0.024°C m$^{-1}$ for a typical thermal conductivity of 2.1 W m$^{-1}$ °C$^{-1}$. As boundary condition at the bed, the temperature gradient

is defined as:

$$\frac{\partial T}{\partial z} = -\frac{g_{hf} + \tau_{b,u} u_b + \tau_{b,v} v_b}{k}$$  (15)

$g_{hf}$ is equal to the geothermal heat flux, $\tau_b$ is the basal shear stress, $u_b$ and $v_b$ are the basal velocities and k is the thermal

conductivity.

### 3.3.2 Surface boundary conditions

As upper boundary condition, the ice temperature of the surface layer is usually set equal to the mean annual air temperature ($T_{ma}$) (Loewe, 1970; Hooke et al., 1983; Zekollari et al., 2017). This is a good approximation for the cold and dry accumulation area (the part of the accumulation area without substantial melt). However, the $T_{ma}$ has been proven to be a major underestimate

for the infiltration recongelation zone, which is the part of the accumulation area where snowmelt occurs (Maohuan et al., 1982, 1990; Arkhipov et al., 2004; van Pelt et al., 2016; Zekollari et al., 2017; Kronenberg et al., 2020). Latent heat released through water percolation and refreezing in firn induces warming (Maohuan, 1990; Huybrechts et al., 1991; van Pelt et al., 2016; Zekollari et al., 2017). Heat conduction and advection expands this warm signal which is often called cryo-hydrological warming (Lüthi et al., 2015). It is important to take this effect into account which raises the temperatures, relaxing associated

viscosity and hence increasing local flow velocities (Colgan et al., 2015). To consider the effect of meltwater refreezing and surface warming, a parametrisation is applied that warms the surface layer for the locations where meltwater is retained at the end of a mass balance year (Reeh, 1991; Huybrechts et al., 1991; Zekollari et al., 2017). Below the equilibrium line, this latent heat is lost together with meltwater runoff over the impermeable ice surface.





However, besides a warming due to refreezing meltwater, other processes play a role in determining the ice temperature at the surface. The presence or absence of snow has been proven to be crucial for the surface temperatures (Hooke et al., 1983; Zhang et al., 2005; Meierbachtol et al., 2015). The most pronounced effect of snow on the surface temperature of the ice mass occurs in the winter. During the winter months, because of the lower thermal conductivity of snow compared to ice, a significant snow cover insulates the ice below as a blanket from the strongest winter cold penetrating in the ice (Hooke et al., 1983; Zhang

et al., 2005; Kronenberg et al., 2020). This ensures that average temperatures below the snow cover are usually much higher compared to the average air temperatures above the snow. For instance, Zhang (2005) reported an increase of the mean annual surface temperature of permafrost of 0.1°C cm$^{-1}$ of maximum snow depth.

In addition to the effect of the insulating snow layer, in the ablation area, the vertical velocity is positive (upward) implying

that ice from deeper layers is advected upwards. Further, the upper ice layers are removed due to melting. Both these processes ensure that the ice temperature at the surface in the ablation area is strongly influenced by the ice temperature of the deeper layers (upward vertical advection). For this reason, Wohlleben et al. 2009 and Blatter et al. 1987 proposed to use the ice temperature gradient to prescribe the ice temperature at the surface as boundary condition. However, this still requires a prescription of the air temperature that influences the ice surface temperature from above.


A last effect that can influence the ice surface temperatures is the flow of water in supraglacial melt channels and deep-water percolation through crevasses and moulins which can provide heat of friction contributing to ice temperature warming in the ablation area (Philips et al., 2010; Gilbert et al., 2020). These processes are not considered in this study because neither for the Grigoriev nor for the Sary-Tor glacier significantly large areas with crevasses or moulins were detected. Observations show

that melt water on both ice masses is mainly discharged supraglacially.

Different methods have been used to determine the surface boundary condition accounting for snow cover and mean July temperatures for the ablation area and the percolation zone (Hooke et al., 1983), or by solely using a firn warming effect in the accumulation area (Zekollari et al., 2017). Here, we apply a combination of both, a simple parametrisation with two tuning

parameters implicitly accounting for all the effects described above:

$$T_s = \begin{cases} \min \left(0°C,\ T_{ma} + w_f r_{rem} + i_s s_{max}\right) & H > ELA \\ \min \left(0°C,\ T_{ma} + i_s s_{max} + m_s \frac{T_{20\,m} - T_{10\,m}}{10}\right) & H < ELA \end{cases} \tag{16}$$

$T_s$ is the ice temperature at the surface which is assumed to be not influenced by the seasonal cycle. Hence, it corresponds

typically to the ice temperature at 10-15 m depth. $T_s$ is used as the upper boundary condition in the model. $T_{ma}$ is the mean annual air temperature, which is calculated by taking the average values of all hours in one year. For the lapse rate, a sinusoidal



function is used with a limit of -0.0065°C m$^{-1}$ in June and -0.0023°C m$^{-1}$ in January, based on data from Aizen et al. (1995). For the calculation of $T_{ma}$, as for the mass balance model, temperature data from the Kumtor-Tien Shan meteorological station (Fig. 1) is used (see Van Tricht et al. (2021b) for more information about this meteorological station). $w_f$ corresponds to the

firn warming parameter (in °C (m w.e.)$^{-1}$) which determines the warming for every metre of refrozen meltwater remaining ($r_{rem}$) at the end of each mass balance year. The $i_s$ parameter (in °C (m w.e.)$^{-1}$) corresponds to the snow insulation effect based on the maximum snow depth at the end of winter ($s_{max}$). When the ice temperature at the surface reaches the melting temperature, the ice temperature is kept constant at 0°C. The temperature gradient between 10-20 m ice depth (($T_{20m} - T_{10m}$)/10) and the surface mass balance ($m_s$) are used to correct for the effect of surface layer removal in the ablation area.


Both the $w_f$ and $i_s$ parameters are calibrated based on historical and spatially distributed ice temperature measurement at 10-15 m depth (assumed to reflect the average annual temperature) on the Grigoriev ice cap (Dikikh et al., 1965; Thompson et al., 1993; Arkhipov et al., 2004; Takeuchi et al., 2014). The calibrated mass balance model (see sect. 3.4) is used to estimate the average amount of $r_{rem}$ at the end of the year, and $s_{max}$ at the end of winter which is typically in May in the area. An overview

of the temperature measurements used for calibration is given in Table 1. The 15 years in advance of each individual measurements of Table 1 are used for the average values of $T_{ma}$, $r_{rem}$ and $s_{max}$. This period is chosen because it is large enough to filter out interannual variability and at the same time to consider sufficient fluctuations in the conditions at the surface.

As can be seen in Table 1, the ice temperatures at 10-15 m depth increase with altitude, while the mean annual air temperatures

decrease with altitude (Table 1). This difference can be explained by the effect of an increasing warming of the surface layer due to refreezing meltwater and maximum snow thickness for insulation.

**Table 1. Measured ice temperatures of the Grigoriev ice cap at 10-15 m depth from 1962 (Dikikh et al., 1965), 1990 (Thompson et al., 1990), 2001 (Arkhipov et al., 2004) and 2007 (Takeuchi et al., 2014). $T_{ma}$ represents the mean annual air temperature at the**

**considered surface elevation of the 15 years in advance of each individual measurement.**

| Year | Height (m a.s.l.) | $T_{10-15\,m}$ (°C) | $T_{ma}$ (°C) |
|------|-------------------|---------------------|---------------|
| 1962 | 4250 | -6.7 | -9.9 |
| 1962 | 4293 | -6.2 | -10.0 |
| 1962 | 4305 | -6.2 | -10.1 |
| 1962 | 4420 | -5.1 | -10.4 |
| 1990 | 4600 | -4.0 | -10.9 |
| 2001 | 4600 | -2.7 | -10.8 |
| 2007 | 4600 | -2.6 | -10.7 |





### 3.4 Mass balance model

To obtain $m_s$, $s_{max}$, and $r_{rem}$, we apply the surface energy mass balance model that has been optimised, calibrated and used in Van Tricht et al. (2021b). The temporal resolution of the mass balance model is 1 hour, while the coupling with the ice flow model occurs yearly. The input data of the mass balance model are limited to a digital elevation model (DEM), air temperatures and solid precipitation at hourly intervals. For meteorological data, the reconstructed data series of the Kumtor-Tien Shan meteorological station is used (Van Tricht et al. 2021b). The DEM is used to calculate the hourly (solar) insolation with respect

to the slope and aspect of every grid point, and the effect of shadow, needed for the energy balance, and to estimate the temperature and precipitation based on the absolute elevation of the grid point.

For the Sary-Tor glacier, we directly use the calibrated parameters from Van Tricht et al. (2021b). However, incorporating the mass balance measurements from 2020/21 in the calibration procedure reveals an optimal $c_0$-value of -199 which is slightly

lower than the obtained value of -193 based on calibration data from 2014-2020. Concerning the Grigoriev ice cap, initially, the same parameters are used as for the Sary-Tor glacier. The $c_1$ parameter is fixed at 24 as this was found to be optimal for the other glaciers in the area (Van Tricht et al., 2021b). However, the $c_0$ parameter as well as the precipitation gradient are then tuned by minimising the root mean square error (RMSE) between modelled and measured mass balances from Dyurgerov (2002), Arkhipov et al. (2004) and Fuijita et al. (2011) (Table 2). For instance, Arkhipov et al. (2004) determined the average

accumulation at the summit of the Grigoriev ice cap based on the detection of annual layers to be 0.26 m w.e. yr$^{-1}$ between 1990 and 2000 which appeared to be slightly less than the 0.32 m w.e. yr$^{-1}$ between 1963 and 1989.

**Table 2. Data used for the calibration of the mass balance model for the Grigoriev ice cap. $m_s$ represent the surface mass balance. All values are in m w.e. yr$^{-1}$.**


| | $m_s$ at the summit | Mean specific $m_s$ | $m_s$ at the front |
|---|---|---|---|
| Average 1963-1989 (Arkhipov et al., 2004) | 0.32 | | |
| 1986-1987 (Dyurgerov, 2002) | | -0.22 | -0.50 |
| 1987-1988 (Dyurgerov, 2002) | | -0.30 | -0.90 |
| Average 1990-2000 (Arkhipov et al., 2004) | 0.26 | | |
| 2005-2006 (Fuijita et al., 2011) | 0.16 | | |
| 2006-2007 (Fuijita et al., 2011) | 0.33 | -0.36 | -1.80 |

A crucial output of the mass balance model for the determination of the thermal regime is the amount of meltwater that remains in the snowpack and refreezes at the end of the mass balance year (see sect. 3.3). Following Van Tricht et al. (2021b), we

assume all snowmelt to be retained in the snowpack until a maximum amount of $P_{max}$ (taken as 60% of the snow depth) is reached (Reeh, 1991). At every timestep, the snow depth and the amount of retained water are updated. Finally, at the end of





the mass balance year, the remaining retained water refreezes and forms $r_{rem}$. The maximum snow depth $s_{max}$ is determined by considering the snow depth at the end of May.

## 3.5 Geometric data

Several geometric datasets are needed to apply the ice dynamic model and the surface mass balance model. The experiments in this study are performed under average 1960-1990 climatic conditions. Therefore, the geometry of the Grigoriev ice cap and the Sary-Tor glacier is reconstructed for 1990.

### 3.5.1 Outlines

The outlines of both ice masses are initially taken from the Randolph Glacier Inventory v6 (RGI, 2017). Subsequently, optical
satellite images from Landsat 5, valid for August 1990, are used to update the outline using manual delineation based on visual inspection of the ice – bedrock boundary.

### 3.5.2 Bedrock DEM

Concerning the Sary-Tor glacier, the ice thickness measurements were performed with a VIRL-6 GPR radar at 20 MHz central frequency (Petrakov et al. 2014). A maximum part of the accessible area of the glacier was covered with a grid of longitudinal
and transverse profiles. Then, an ice thickness field was reconstructed using interpolation with an estimated accuracy of 15-20 m and a volume of $0.126 \pm 0.001$ km$^3$ was obtained (Petrakov et al., 2014). A bedrock DEM is obtained by subtracting the ice thickness of Sary-Tor glacier in 2013 from the surface elevation in 2013 (obtained from the TanDEM-X DEM).

Concerning the ice thickness of the Grigoriev ice cap, we performed ~ 500 GPR-measurements over the entire ice cap in the
summer of 2021 using a Narod Radio Echo Sounding system with a lower frequency of 5 MHz. Subsequently, following Van Tricht et al. (2021a), a glacier wide ice thickness field is reconstructed using a constant yield stress model for the unmeasured areas. During the field campaign, UAV images were collected using a DJI Phantom 4 RTK from about 200 m above the ice cap. These images were subsequently used in Pix4D to infer an accurate and centimetre resolution digital surface model (DSM), as in Van Tricht et al. (2021c). A bedrock DEM is determined by subtracting the reconstructed ice thickness
distribution from the DSM.

### 3.5.3 DEM of 1990

For consistency with the 1960-1990 climatic conditions, a DEM is created for 1990 as initial state to be used in the glacier model. Using the calibrated mass balance model (see sect. 3.4), the cumulative mass balance is calculated between 1990 and the year in which the ice thickness was measured (2021 for the Grigoriev ice cap and 2013 for the Sary-Tor glacier). Under
the assumption that ΔSMB = ΔH (hence neglecting the role of ice dynamics for these periods), the surface elevation of 1990



is then inferred by adding ΔSMB to the DEM of 2021 and 2013 for the glacial area (Fig. 2). Outside the glacial area of 2013 and 2021, the surface elevation is reconstructed by setting the elevation equal to the bedrock elevation at the 1990 front, attaching to the DEM of 2013/2021 at the observed glacier front, an interpolating in between (Fig. 2).


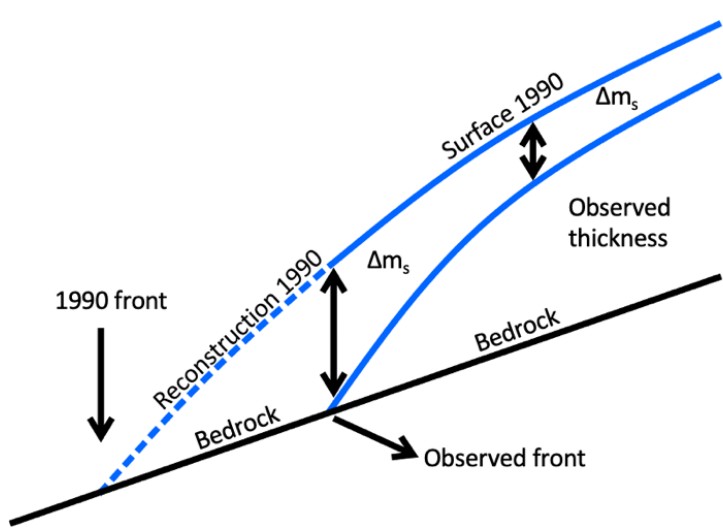

**Figure 2: Reconstruction of the 1990 geometry from a combination of modelling the cumulative surface mass balance (Δm$_s$) and interpolation.**

**3.5.4 Little Ice Age mask**

A LIA ice mask is created from visual detection of unique LIA moraines, clearly present on satellite images (Landsat and Sentinel-2) for both the Grigoriev ice cap and the Sary-Tor glacier. This ice mask serves as the maximum area of both ice masses which is allowed to be occupied by ice. When the ice were to expand beyond the mask, the ice is removed. However, this never applies to the geometry for average 1960-1990 climatic conditions as this remains entirely within the LIA mask.

**3.6 Experiments and sensitivity analysis**

First, multiple experiments are performed using the coupled surface mass balance and the ice flow models with standard settings for the basal sliding parameter A$_{SL}$ (5.10$^{-14}$ m$^8$ N$^{-3}$ yr$^{-1}$). The velocity and temperature fields are run under constant 1960-1990 climatic conditions. During the runs, the geometry is freely evolving with a maximum limit of the LIA extent, until a steady state is reached. When a steady state is reached, modelled ice thickness is compared with the 1990 reconstructed
thickness, and temperature fields are compared with measured temperatures as well as with detected boundaries between cold and temperate ice (Petrakov et al., 2014). A constant mass balance correction is implemented to reach the extent of the 1990 geometry.





Then, a sensitivity analysis is performed with regards to the different parameters affecting the temperature distribution in the
ice masses. This concerns specifically the $i_s$ and $w_f$ parameters, which are altered within ±10%. In each run, one parameter is
changed while the other remains fixed at its optimal value. The sensitivity with regard to the geothermal heat flux is also
assessed by using 50% and 200% of the initial $g_{hf}$-value. In every experiment, first, the geometry is freely evolving until a
steady state is reached using the optimal parameter values. Then, the parameter under consideration is altered, and the geometry
is kept fixed to avoid an effect of a thinner or thicker ice mass caused by a change in the ice viscosity and to simplify the
interpretation of the results.

Finally, a sensitivity to changes in boundary conditions is investigated by altering the precipitation {-40% to 40%} and the
surface temperatures {-2°C to +3°C}. These ranges were selected based on observed (Van Tricht et al., 2021b) and projected
changes. For this analysis, the geometry can evolve freely until a new steady state is reached in equilibrium with the altered
surface boundary conditions. This analysis can give an indication about the evolution of the thermal regime in the past and
into the future, which is part of the discussion section.

## 4 Results

### 4.1 Geometric input data and characteristics

During the 2021 field campaign, ~500 well-spread ice thickness measurements were performed on the Grigoriev ice cap,
reducing the necessity for interpolation over larger distances between the measurement transects (Fig. 3). In 2021, the
Grigoriev ice cap is characterised by an area of 7.545 km$^2$, a maximum ice thickness of 114 m, and a total ice volume of 0.391
km$^3$. Regarding the Sary-Tor glacier in 2021, the glacier area amounts to 2.250 km$^2$ with a maximum ice thickness of 157 m
and a total ice volume 0.118 km$^3$, based on measurements of Petrakov et al. (2014) which we corrected for melt between 2013
and 2021.

Reconstructed for the 1990 geometry, we obtain an area of the Grigoriev ice cap of 8.746 km$^2$ (~15% larger), with a maximum
ice thickness of 115 m, and a total ice volume of 0.513 km$^3$ (~30% more). The corresponding values for the Sary-Tor glacier
are 2.789 km$^2$ (~21% larger), a maximum ice thickness of 158 m, and a total volume of 0.158 km$^3$ (~34% more). Hence, both
ice bodies lost a substantial fraction of their area and volume over the last 31 years, similar to observations in other studies
(Aizen et al., 2006; Kutuzov and Shahgedanova, 2009; Sorg et al., 2012). The created 1990 geometry of both ice masses is
used to initiate the experiments below. The eastern tributary of the Sary-Tor glacier is not included for the 1960-1990 runs, as
it was already disconnected from the Sary-Tor main glacier at the end of this time frame.



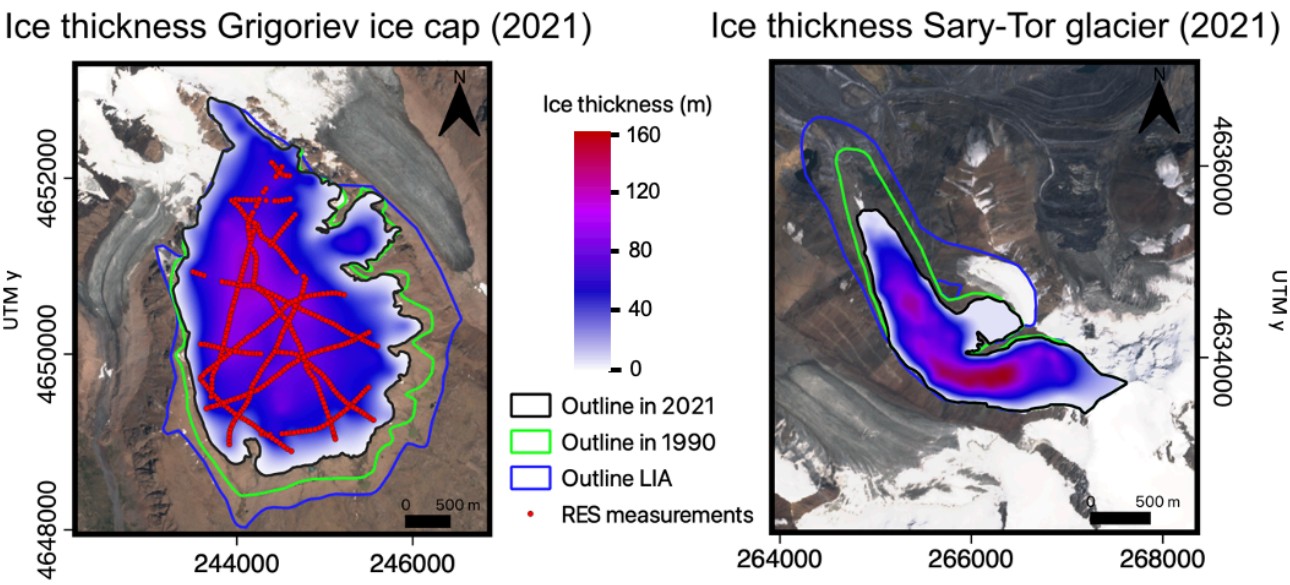

**Figure 3: Ice thickness distribution of the Grigoriev ice cap and the Sary-Tor glacier in 2021. The locations of the ice thickness measurements of 2021 on the Grigoriev ice cap are added in red. The background is a Sentinel-2 image from 26th of July 2021.**

## 4.2 Calibrated parameters and boundary conditions

### 4.2.1 Surface mass balance

The calibration of the surface mass balance model for the Grigoriev ice cap reveals optimal values for a precipitation gradient of +150 mm km$^{-1}$ yr$^{-1}$ and a $c_0$ value of -197 W m$^{-2}$. This rather low value for the precipitation gradient (compared to the Sary-Tor glacier) appears to be necessary to limit the accumulation on the summit of the ice cap, as was measured and reconstructed by Arkhipov et al. (2004) and Fuijita et al. (2011). Such a low gradient can be explained by the location of the Grigoriev ice cap on the top of a mountain, lower than the surrounding mountains, reducing the effect of orographic precipitation enhancement. At the other hand, the Sary-Tor glacier is located at the north-western side of the Ak-Shyirak massive which could be more exposed to wind and precipitation. The calibrated $c_0$ value is of equal magnitude compared to the values found for the nearby located Bordu and Sary-Tor glaciers (Van Tricht et al., 2021b).

Based on the calibrated surface mass balance model, and the average climatic input data of 1960-1990, at similar elevation, the ablation is about 1 m w.e. yr$^{-1}$ more negative for the Grigoriev ice cap compared to the Sary-Tor glacier, which is due to the orientation (mostly south vs northwest) and the lower amount of precipitation. The estimated ELA for the 1960-1990 period is around 4300 m a.s.l. for the Grigoriev ice cap, which is close to the average ELA found in Arkhipov et al. (2004), and about 4200 m a.s.l. for the Sary-Tor glacier. For average 1960-1990 climatic conditions, the mean specific mass balance is +0.15 m



w.e. yr$^{-1}$ for the Grigoriev ice cap, and -0.06 m w.e. yr$^{-1}$ for the Sary-Tor glacier (Fig. 4). For the experiments with the ice flow

model, the surface mass balance of the Sary-Tor glacier is adjusted by adding +0.08 m w.e. yr$^{-1}$, and the surface mass balance of the Grigoriev is adjusted by -0.05 m w.e. yr$^{-1}$ to obtain a steady state with an extent as close as possible to the extent of the reconstructed 1990 geometry (see sect. 4.3).

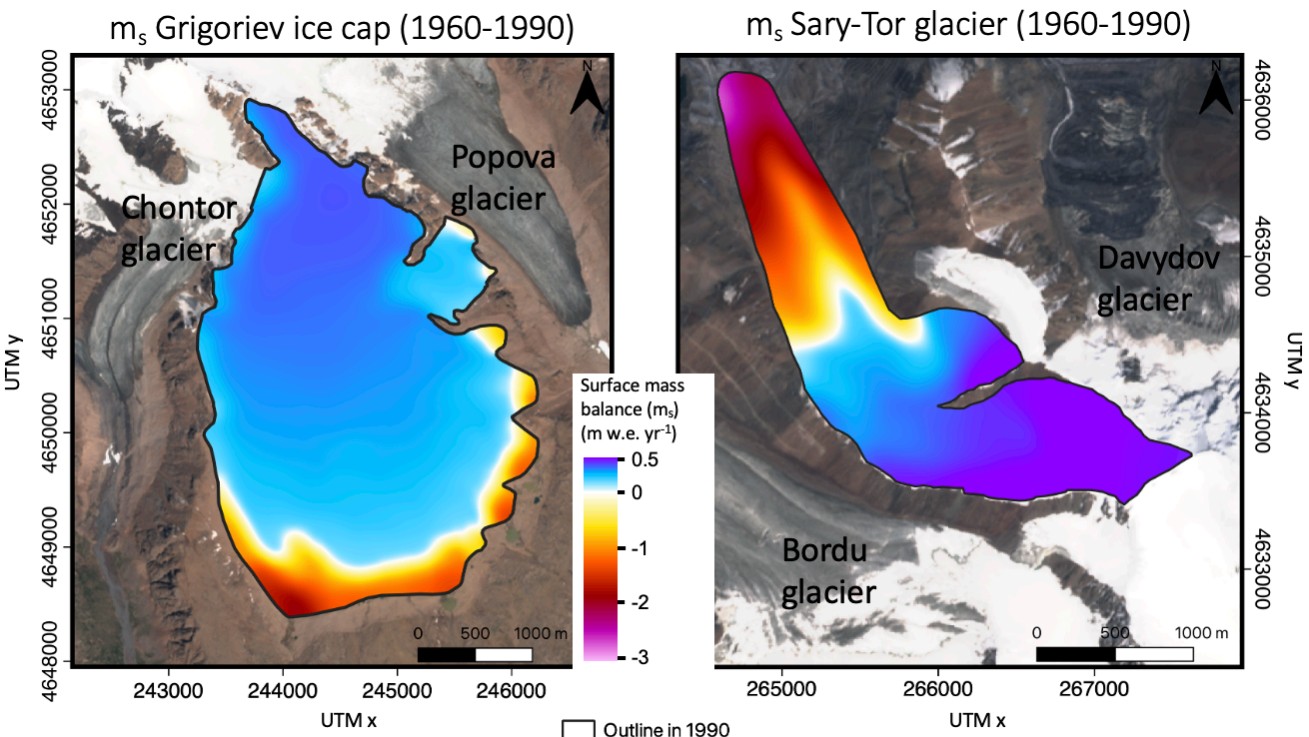

**Figure 4: Average surface mass balance (m$_s$) in m w.e. yr$^1$ of the Grigoriev ice cap and the Sary-Tor glacier for the mean 1960-1990 climatic conditions. The background is a Sentinel-2 image from 26$^{th}$ of July 2021.**

**4.2.2 Firn warming and snow insulation**

Using the output from the surface mass balance model, r$_{rem}$ and s$_{max}$, and the temperature data, the optimal combination of the

460 w$_f$ and the i$_s$ parameters is 41 and 22 (in °C (m w.e.)$^{-1}$) (Fig. 5a). The corresponding RMSE between the measured (see Table 1) and modelled temperatures at 10-15 m depth is equal to 0.28°C (Fig. 5a,b), which is a close correspondence.





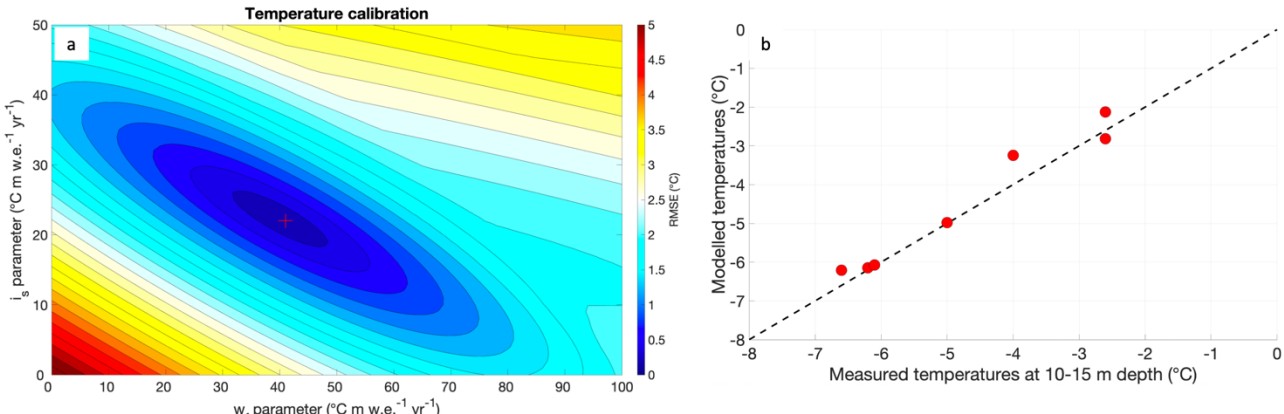

**Figure 5: (a) Calibration of the $w_f$ and the $i_s$ parameters based on ice temperature measurements of the Grigoriev ice cap. (b) Modelled and measured temperatures at 10-15 m depth for different elevations and times of the Grigoriev ice cap.**

The obtained $w_f$ value (41) is larger than values found in previous studies for e.g. the Greenland ice sheet (26.6; Reeh, 1991; Huybrechts et al., 1991), and Hans Tausen ice cap (22.2; Zekollari et al., 2017). However, theoretically, refreezing 1 m$^3$ water (~ 1000 kg) in an ice cube of 5 m$^3$ (which is the size of a typical surface layer when 21 equally spaced layers are considered for an ice thickness of 100 m), would be able to supply a warming of 35-42°C (m w.e.)$^{-1}$ yr$^{-1}$ which is close to the value found in this study (Lüthi et al., 2015).

The parameter to consider for the insulation of snow (optimal $i_s$ = 22) is slightly larger than the values found in Hooke et al. 1983 (about 4°C for 1 metre of snow (~ 12-20°C (m w.e.)$^{-1}$ yr$^{-1}$) on the Storglaciären glacier, or about 2.5°C warming for 0.6 metres of snow (~ 13-21°C (m w.e.)$^{-1}$ yr$^{-1}$) for the Barnes ice cap). However, Zhang (2005) and Yershov (1998) reported a mean annual temperature increase of about 0.1°C cm$^{-1}$ of snow, which is equivalent to 10°C m$^{-1}$ yr$^{-1}$ of snow (~ 30-50°C (m w.e.)$^{-1}$ yr$^{-1}$). Further, values of equal magnitude were also obtained in Liang and Zhou (1993) and Zhou et al. (2000), which increases the confidence in the value derived in this study.

### 4.2.3 Surface warming and temperatures

The input of latent heat from percolating and refreezing meltwater is the strongest in the middle part of the accumulation zones (Fig. 6). Closer to the equilibrium line, the snowpack present at the end of the year is too thin for large amounts of refrozen meltwater. At the highest elevations, the amount of refrozen water is low, caused by the absence of substantial melt (too low temperatures).

The effect related to snow insulation is more homogeneously distributed over the ice bodies because almost all precipitation in winter is snow, independent from the elevation on the glacier, which ensures that differences in maximum snow depth at the end of winter are only related to precipitation enhancement with elevation. Combining the effect of firn warming and snow



insulation, an entirely cold surface is obtained for the Grigoriev ice cap while a polythermal surface is obtained for the Sary-Tor glacier (Fig. 6).

For both ice masses, the coldest ice temperatures at the surface can be found in the ablation areas, where the formation of superimposed ice at the end of summer is absent because the meltwater is evacuated and not retained in the snow, and where a thinner snow cover in winter ensures a larger cooling of the surface (Fig. 6). The temperatures are the highest in the accumulation areas. Concerning the Sary-Tor glacier, there is a temperate band between ~4200-4500 m with temperatures

equal to zero degree. The hypothesis is that these higher ice temperatures are dissipated downwards through conduction and vertical movement towards lower elevations being the reason for the expected polythermal regime of the valley glaciers (Petrakov et al., 2014), such as also observed on the Tuyuksu glacier in the Northern Tien Shan (Nosenko et al., 2016). At the highest elevations of the Sary-Tor glacier, the ice temperature at the surface are again lower because of the absence of melt water available for refreezing.

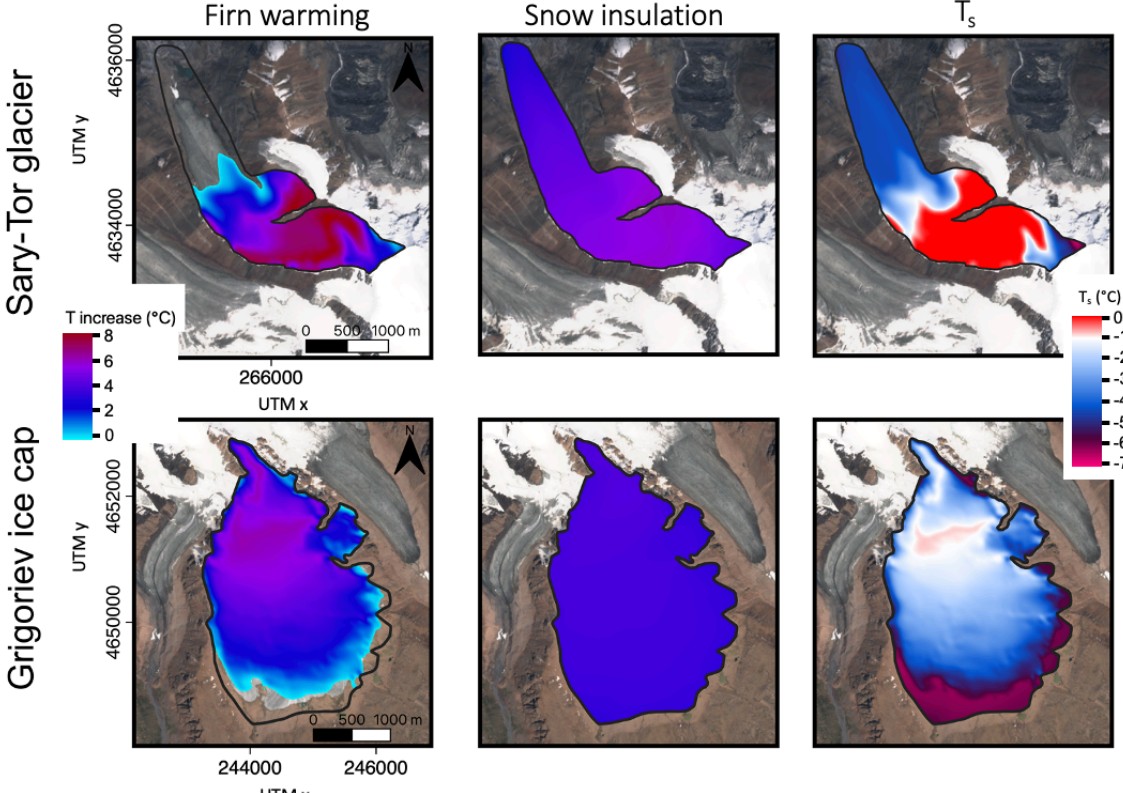

**Figure 6: Warming due to refreezing of meltwater (firn warming) and due to snow insulation and resulting mean annual 10 m ice temperature ($T_s$) for both ice masses according to constant 1960-1990 climatic conditions. The background is a Sentinel-2 image**

**from 26th of July 2021.**



### 4.3 Modelled steady state velocities and thickness

Under constant 1960-1990 climatic conditions and with the applied surface mass balance adjustments, the obtained steady
states match the reconstructed 1990 geometry (extent and ice thickness) relatively well (Fig. 7). The RMSE between the
reconstructed and modelled ice thickness is about 22.01 m for the Sary-Tor glacier and 15.2 m for the Grigoriev ice cap. The
enhancement factor is set at 3 for the Grigoriev ice cap and at 4 for the Sary-Tor glacier, which appears to be the optimal value
to match with the reconstructed ice thickness of 1990.

Even though there is generally a good agreement between modelled and reconstructed ice thickness, a few larger discrepancies
exist (Fig. 7). For the highest parts of the Sary-Tor glacier, the modelled ice thickness is smaller than the reconstructed ice
thickness. However, in this area, no GPR measurements were performed, demonstrating that this area is associated with the
largest uncertainty in ice thickness (Petrakov et al., 2014). In addition, the modelled ice thickness is specifically larger than
the reconstructed ice thickness on the lateral margins of the Sary-Tor glacier in the ablation area. This is likely related to the
520 reconstructed geometry of 1990, or it might be related to the reconstructed geometry not being in equilibrium with the 1960-
1990 climatic conditions (Fig. 7). Furthermore, the modelled ice thickness is thicker for the eastern tributary, but here, as well,
no direct ice thickness measurements were performed (Petrakov et al., 2014).

Regarding the Grigoriev ice cap, the modelled ice thickness is thicker and extents generally further at the edges of the ice cap
compared to the 1990 reconstruction. At the top, on the other hand, the modelled ice thickness is thinner (15-25 m). The latter
might be associated to the complexity of modelling ice thickness in the vicinity of ice divides or to the measurements of the
ice thickness itself. It is unlikely that the underestimated modelled ice thickness at the top of the Grigoriev ice cap is related to
an excessively high ice temperature, as to correctly model the ice thickness here would require an enhancement factor of 1,
which corresponds to a lower temperature of -5 degrees. This is far outside the limit of the values of the observed ice
temperatures (Thompson et al., 1993; Arkhipov et al., 2004; Takeuchi et al., 2014).





**Figure 7: (a,c) Ice thickness difference between the modelled steady state (~ 1960-1990 average climatic conditions) and the reconstructed geometry of 1990. The used flowline is added on the map with a black line. (b,d) Reconstructed (initial glacier surface) and modelled (glacier surface) ice thickness along the central flowlines. The upper panels are for the Sary-Tor glacier, the lower for the Grigoriev ice cap. The background of the left maps is a Sentinel-2 image from 26th of July 2021.**

Regarding the horizontal (surface) ice velocities, the Grigoriev ice cap is generally characterised by a very low velocity between 0 and 5 m yr$^{-1}$ (Fig. 8). Such low values were also observed by Fuijita et al. (2011) and modelled by Nagornov et al. (2006). The very low ice velocities imply that the ice is only advected very slowly downstream, keeping ice fluxes low. Only for the outlet glacier in the eastern part, which flows towards the Popova glacier, velocities increase to > 20 m yr$^{-1}$ due to the steeper slope in this area. The ice velocities are mostly significantly higher for the Sary-Tor glacier with a maximum slightly above 30 m yr$^{-1}$ (Fig. 8). This implies that the horizontal advection of ice for Sary-Tor glacier is much more pronounced.





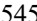

**Figure 8: (a,c) Horizontal surface ice velocities of the Grigoriev ice cap and the Sary-Tor glacier. The flowline is added on the map with a black line. (b,d) Horizontal ice velocities along the lowlines of both ice masses. The upper panels are for the Sary-Tor glacier, the lower for the Grigoriev ice cap. The background of the left maps is a Sentinel-2 image from 26th of July 2021.**

## 4.4 Thermal regime under the 1960-1990 climatic conditions

With the imposed boundary conditions and the modelled thickness and velocity structure, the steady state englacial ice temperatures are determined. They range between -7.70°C and 0°C for the Sary-Tor glacier and between -6.61°C and -0.03°C for the Grigoriev ice cap. A polythermal regime is obtained for the Sary-Tor glacier (Fig. 9). Temperate ice is present over a

555 large part of the glacier base, originating in the accumulation area which is subsequently advected downstream with the ice



flow. At the snout of the Sary-Tor glacier, and in the largest part of the ablation area, the ice is cold. Over the entire glacier volume, about 38% of the ice of the Sary-Tor glacier is temperate, classifying this glacier reliably as a polythermal glacier. Concerning the Grigoriev ice cap, the model results show cold ice except for a very thin layer near the bed in the central part, which is characterised by temperate ice. This concerns less than 1% of the total volume. Hence, the Grigoriev ice cap can be classified as a cold ice cap. The thin layer of temperate ice in the central area was also observed on GPR profiles measured by I. Lavrentiev (personal communication).

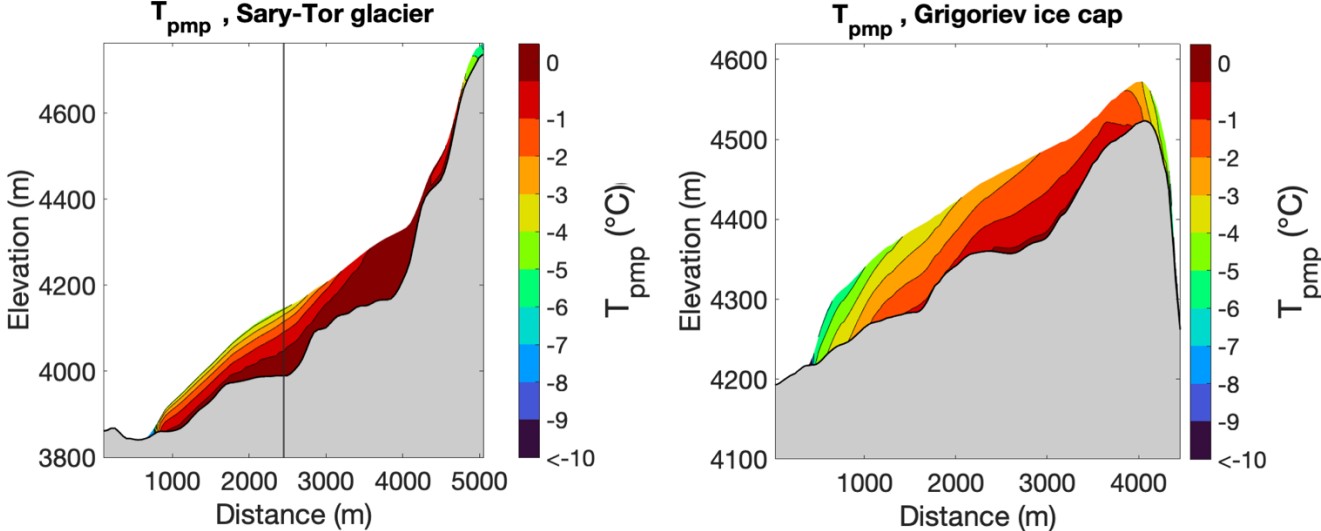

**Figure 9: Modelled ice temperatures of the Sary-Tor glacier and the Grigoriev ice cap along the central flowline using a free evolving geometry and corrected for the pressure melting point. The vertical line in the plot of the Sary-Tor glacier corresponds to the cross-section indicated on Fig. 10.**

The difference between the thermal regime of the two ice masses can be attributed to a combination of several factors. These are related to the surface conditions, the ice thickness, and the ice dynamics. First, the accumulation area of the Grigoriev is located at higher altitudes and has therefore a lower $T_{ma}$. Next to that, the Grigoriev ice cap is characterised by a smaller amount of $s_{max}$ and $r_{rem}$ reducing the firn warming effect and the effect of insulating snow. Further, the ice thickness of the Grigoriev ice cap is thinner, resulting in an improved transport of the geothermal heat flux. Finally, the faster ice flow of the Sary-Tor glacier leads to significantly more efficient advection that can transport the heat produced in the lower accumulation area downstream, which is absent for the Grigoriev ice cap.

The difference between the thermal regime of the Grigoriev ice cap and the Sary-Tor glacier found in this study ensures that both ice masses are characterised by different dynamics and characteristics. Because of the lower average ice temperature, the





ice viscosity of the Grigoriev ice cap is larger than that of the Sary-Tor glacier. This ensures a greater stiffness which causes
the ice intrinsically slower to deform. Further, the presence of temperate basal ice for the Sary-Tor glacier implies that this ice
mass can slide over its base, while this is not the case for the Grigoriev ice cap. Using a typical basal sliding parameter (see
sect. 3.6), a maximum basal sliding speed of 13 m yr$^{-1}$ is found for the Sary-Tor glacier, for which 55% of the bed is at the
pressure melting point. Such basal sliding was also observed for the Ürümqi glacier in the Chinese Tien Shan (Maohuan et al.,
1989,1992). In addition, the presence of temperate ice at the surface causes the Sary-Tor glacier to have zones around the
elevation of the EL where water can infiltrate in the glacier (Fig. 9). In the ablation area of the Sary-Tor glacier, all the water
drains away superficially, which can be explained by the impermeable cold ice (Fig. 9). This is the case for the entire surface
of the Grigoriev ice cap. As a consequence of the ice temperature differences, it might be more difficult to reproduce measured
flow velocities correctly with ice flow models employing isothermal (cold or temperate) ice (Riesen et al., 2010; Ryser et al.,
2013).

### 4.4.1 Validation of the thermal regime of the Sary-Tor glacier


While the ice temperature measurements of the Grigoriev ice cap are used to calibrate the firn warming and snow insulation
parameters (see sect. 4.2.2), a validation of the thermal regime of the Sary-Tor glacier is performed by comparing modelled
ice temperatures with GPR profiles of Petrakov et al. (2014) showing the boundary between temperate and cold ice. For this,
a transverse profile in the middle part of the glacier is considered (Fig. 9). There is clearly a similarity regarding the presence
of temperate ice over the bedrock, which is about 1/3 of the total ice thickness (black line) (Fig. 10).

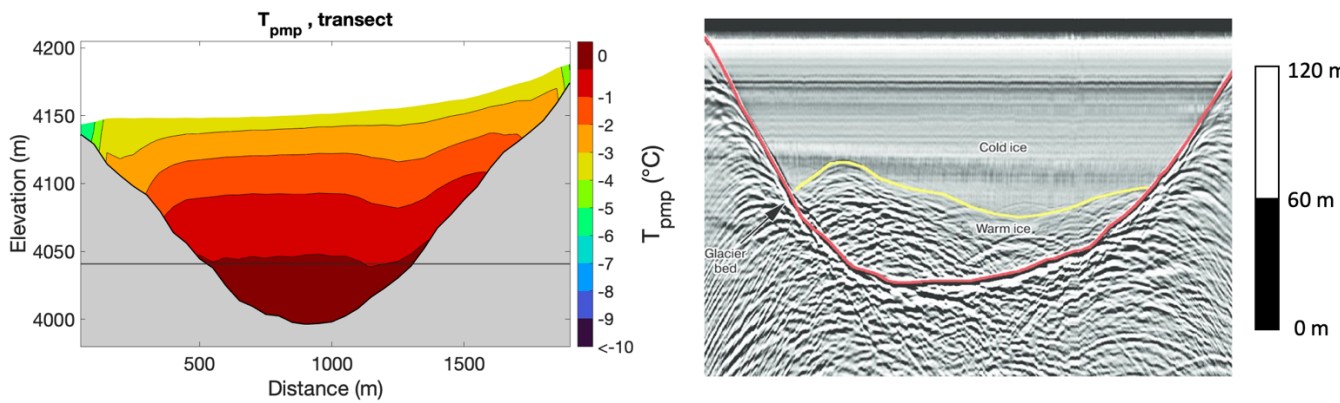

**Figure 10: Validation of the thermal regime with GPR profiles showing the boundary between cold and temperate ice. The black**
**line in the left figure shows the ratio of the thickness of the temperate layer to the total thickness in the right figure, which is about**
**one-third. The right-hand figure is reproduced from Petrakov et al. (2014) with permission from the publisher.**





In addition, a point-by-point comparison with measurements of Petrakov et al. (2014) is performed which reveals a mean difference of -2 m i.e., and a root mean square error of 28 m i.e. between modelled and measured thickness of the temperate ice layer. In addition, for all locations where temperate ice was effectively observed on the GPR profiles, the model output also shows the presence of temperate ice. Given the uncertainty in both the measurements and the model output, this is a strong

agreement, and it increases the confidence in the obtained results.

### 4.4.2 Ice temperature profile at the summit of the Grigoriev ice cap

For the Grigoriev ice cap, the modelled temperature profile for constant 1960-1990 climatic conditions is compared with the measured ice temperatures of Takeuchi et al. 2014 from 2007 near the summit of the ice cap (Fig. 11). A large mismatch is found for depths below 20 metres which increases with depth. Since the temperatures of the measured profile decrease with

depth, while there cannot be advection of cold ice from upper areas close to the summit, this must be related to the recent warming which has not yet reached the lower ice layers. In other words, the measured ice temperature profile of 2007 is not in equilibrium with the imposed average climatic conditions. With very limited accumulation rates (typically 0.3 m w.e. $yr^{-1}$), englacial ice temperatures react slowly to temperature changes at the surface.

For a bottom temperature of -4°C (as was measured by Takeuchi et al. 2014), the surface temperatures should have been at least 2-3 degrees colder. Such a temperature increase was noticed by Thompson et al. (1993) by comparing borehole temperatures at 10-15 m depth of the sixties (Dikikh, 1965) with 1990 measurements on the Grigoriev ice cap. However, as no such temperature increase is recorded at the nearby Kumtor-Tien Shan weather station nor a significant increase in precipitation (Van Tricht et al., 2021b), this rapid increase in ice temperature is likely related due to an increase in the formation

of $r_{rem}$ or because of a lowering of the surface.

To assess the time varying ice temperature for the location of the borehole, the surface ice temperature is calculated for the 1920-1950 period and the 1980-2010. The latter is about 4 degrees lower for the 1920-1950 mean climatic conditions. The difference is caused by a $T_{ma}$ difference of 1.1°C (recorded at the Kumtor meteorological station), a difference of 2.5°C caused

by an increase in $r_{rem}$, and a difference of 0.4°C related to a change in snow insulation. By gradually increasing the surface temperatures between 1920-1950 and 1980-2010 (0.06°C $yr^{-1}$) for a time-dependent temperature profile, the mismatch between modelled and measured temperatures is strongly reduced (yellow curve in Fig. 11). The above analysis also demonstrates clearly that the largest contributor to the ice temperature increase is the strong increase in $r_{rem}$.



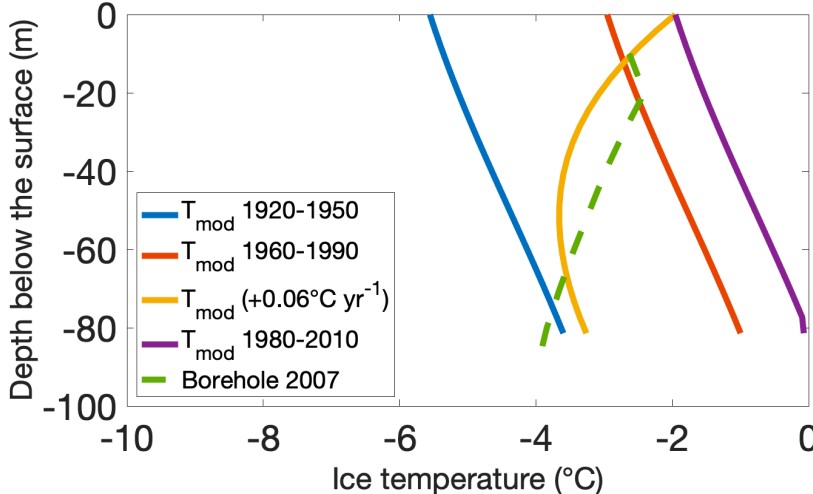

**Figure 11: Measured (2007) and modelled temperature profile at the summit of the Grigoriev ice cap for a fixed geometry. The steady state temperature profile is modelled for the 1920-1950, 1960-1990, and the 1980-2010 mean climatic conditions. A time-dependent temperature profile is obtained for 1980-2010 by increasing the surface temperature from 1920-1950 with 0.06°C yr$^{-1}$.**

## 5 Sensitivity study

### 5.1 Uncertainty and sensitivity to parameters

The sensitivity experiments bring to light that the fractional amount of temperate ice varies between 34-47% for the Sary-Tor glacier, and between 0 and 3% for the Grigoriev ice cap depending on the choice of the parameters associated with the firn warming, the insulation of snow, and the magnitude of the geothermal heat flux (Table 3). Hence, the fractional amount of temperate ice can vary significantly, but in every case, there exists a substantial amount of temperate ice for the Sary-Tor glacier and very little to none for the Grigoriev ice cap. Further, and in general, we find a lower sensitivity of the results for the Grigoriev ice cap, which can mainly be attributed to the lower amounts of $s_{max}$ and $r_{rem}$ for the ice cap and its lower average temperature.

For both ice bodies, the average ice temperature is most sensitive to changes of the geothermal heat flux, while the fractional amount of temperate ice changes only slightly when the geothermal heat flux is reduced to 50% or increased to 200% (Table 3). Hence, based on the sensitivity analysis, it can be concluded that for the selected ice masses, parameter uncertainty and the magnitude of the geothermal heat flux do not have a major influence on the obtained thermal regime, making the results of this study robust.



Other assumptions that influence the results of this study concern the air temperature lapse rates (based on Aizen et al., 1995) and the precipitation gradients. The impact of the selected lapse rates is likely small because its uncertainty is implicitly taken into account in the calibration of the $w_f$ and $i_s$ parameter (through obtaining $T_{ma}$). Concerning the precipitation gradient, a lower gradient (compared to the Sary-Tor glacier, this differs by 100 mm at 4600 m altitude) proved to be necessary to obtain

a sufficiently low accumulation on the summit of the Grigoriev ice cap. In the case of a larger precipitation gradient, a similar one as for the Sary-Tor glacier, $s_{max}$ and $r_{rem}$ increase. Using these higher values would result in a slightly lower value of $w_f$ and $i_s$. Nevertheless, even with the use of a lower $w_f$ and $i_s$ value, a polythermal regime is obtained for the Sary-Tor glacier and a cold regime for the Grigoriev ice cap (Table 3). As such, the precipitation gradient also has a small influence on the results achieved.


Other uncertainties that contribute to the results found in this study concern uncertainties in the mass balance measurements used for the calibration of $c_0$ for the Grigoriev ice cap, and the measured ice temperatures used for the calibration of $w_f$ and $i_s$. However, the calibration of these parameters has been optimised by selecting all available independent borehole ice temperature and mass balance measurements. Due to the large amount of data available in time and space (see Table 1 and

Table 2), the impact of inaccurate measurements is minimised, and the values found are reliable. Finally, all experiments in this study are performed under constant mean 1960-1990 climatic conditions. However, in reality, warmer and colder years, drier and wetter years, alternate, and the equilibrium line moves up and down. The values of $s_{max}$ and $r_{rem}$ vary accordingly. These interannual variations ensure that the transitions in surface ice temperature are in reality more gradual than found in this study.


**Table 3: Change in average temperature and percentage of temperate ice of both ice masses based on parameter changes of ±10% for the firn warming ($w_f$) and the snow insulation ($i_s$), and 50% vs 200% of the geothermal heating ($g_{hf}$).**

| | Grigoriev ice cap | | | Sary-Tor glacier | | |
|---|---|---|---|---|---|---|
| | Reduction | | Increase | Reduction | | Increase |
| $w_f$ | 0% | | 2% | 35% | | 42% |
| | -3.4°C | Initial values | -2.8°C | -1.8°C | Initial values | -1.6°C |
| $i_s$ | 0% | 1% | 2% | 34% | -1.7°C | 41% |
| | -3.4°C | -3.1°C | -2.7°C | -1.9°C | 38% | -1.5°C |
| $g_{hf}$ | 0% | | 3% | 37% | | 39% |
| | -3.5°C | | -2.5°C | -1.7°C | | -1.5°C |





**5.2 Thermal regime since the LIA and into the future**

A glacier/ice cap thermal regime responds to climate change because the surface boundary condition strongly depends on air temperature and precipitation and because the ice mass geometry adjusts itself. To assess the evolution of the thermal regime of the Grigoriev ice cap and the Sary-Tor glacier, an analysis is performed to examine the effect of changing climatic conditions on the thermal regime of both ice masses by altering the air temperatures (between -2 and +3°C) and precipitation (between -

40 and +40%). This range was selected based on historical temperature and precipitation records at the Kumtor – Tien Shan weather station (Van Tricht et al., 2021b), and expected changes in the future. The retreat or expansion of both ice masses is considered by letting the geometry freely evolve until a new steady state is reached, in equilibrium with the adjusted climate.

For a temperature increase of 2-3°C, the Sary-Tor glacier retreats to high elevations with a remaining volume which becomes

significantly smaller. When the amount of precipitation remains constant or increases only slightly, which is suggested by most climate models (Sorg et al., 2012; Shahgedanova et al., 2020), the fractional amount of temperate ice becomes smaller (Fig. 12). However, when the temperature increase is accompanied by substantial increase in precipitation (+40%), slightly more ice remains, and the ratio becomes larger again. This can be attributed to increased snow insulation and a thicker snow layer in which meltwater can be retained. For colder climates, the Sary-Tor glacier would most likely have been polythermal

as well (Fig. 12). Only for a very strong decrease in temperature (-2°C) and a strong decrease in precipitation (-40%) does the Sary-Tor glacier become significantly colder with a fractional amount of temperate ice smaller than 5%. However, this climate configuration is rather unlikely as was shown by the climate reconstruction in Van Tricht et al. (2021b). Therefore, it can be concluded that the Sary-Tor glacier can always be considered a polythermal glacier since the LIA and into the future, until it disappears as soon as temperatures rise too much.


Concerning the Grigoriev ice cap, the fractional amount of temperate ice remains low in all scenarios except for a strong increase in precipitation (Fig. 12). For instance, increasing the amount of precipitation with 40% while keeping the temperatures fixed at the 1960-1990 average results in a transition from a cold to a polythermal ice cap. This can be explained by a larger insulation by snow and a larger amount of $r_{rem}$ formation. For increasing temperatures, the Grigoriev ice cap retreats

strongly (it disappears when temperatures rise between 2-3°C for equal precipitation) and it remains cold. For lower temperatures, the Grigoriev ice cap expands, but it remains cold as well. Hence, the Grigoriev ice cap can be regarded as a cold ice cap over time.



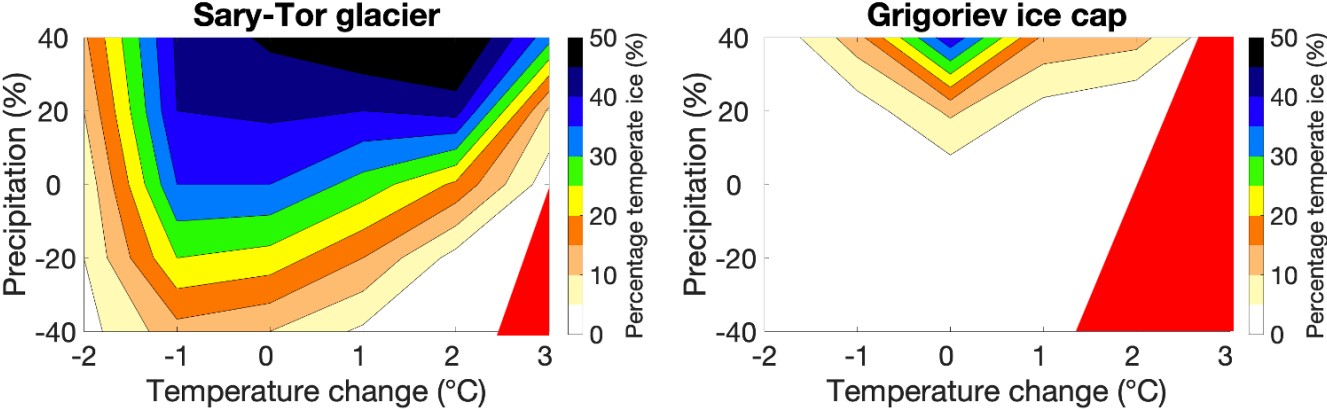


**Figure 12: Fractional amount of temperate ice for different combinations of temperature and precipitation changes. For combinations in the red area, the ice mass disappears completely.**

As demonstrated, the thermal regime is not a constant and ice temperatures can vary substantially over time. Ice temperatures at higher altitudes, where melting is currently limited by the colder climate, rise stronger than air temperatures, because of the enhanced formation of superimposed ice. Further, most climate models indicate a slight increase in winter precipitation (Sorg et al., 2012), which would imply an increase of $s_{max}$ and thus a strengthening of the snow insulation effect. This has already been observed at the Abramov glacier in the southwestern Tien Shan (Kronenberg et al., 2020). As a result, englacial

temperatures generally rise, and part of the glaciers or ice caps in the Inner Tien Shan can undergo a transition from a cold to a polythermal or from a polythermal to a temperate regime (Arkhipov et al., 2004; Gusmeroli et al., 2012). Such a transition enhances ice flow and can fundamentally alter the ice dynamics (Marshall, 2021).

## 6 Conclusion

In this study, we investigated the thermal regime of the Grigoriev ice cap and the Sary-Tor glacier, both located in the Inner

Tien Shan in Kyrgyzstan (Central Asia), using a time-dependent 2-dimensional surface energy mass balance model coupled to a 3-dimensional higher-order thermomechanical ice flow model. Both models were calibrated with mass balance observations and temperature measurements for different periods and elevations. After calibration with the time-dependent output from the mass balance model, a firn warming parameter of 41°C (m w.e.)$^{-1}$ yr$^{-1}$ and a snow insulation parameter of 22°C (m w.e.)$^{-1}$ yr$^{-1}$ was found. Using these parameters, the ice temperature at 10-15 m depth of the Grigoriev ice cap could

be modelled with a root mean square error of 0.28°C at locations with observations. The calibrated parameters were then used to model the thermal regime of both ice masses for constant 1960-1990 climatic conditions.

The simulations revealed a polythermal structure of the Sary-Tor glacier and a cold structure of the Grigoriev ice cap, apart from a thin layer of temperate ice near the bed in the central part. The difference can be attributed to a combination of a smaller amount of maximum snow depth at the end of winter, reduced formation of refrozen meltwater at the end of summer, as well as lower ice flow velocities and thinner ice of the Grigoriev ice cap. As a result, the Sary-Tor glacier is characterised by basal sliding and temperate ice in the upper layers where water is able to infiltrate, while this is not the case for the Grigoriev ice cap. An extensive sensitivity analysis was performed to analyse the thermal structure for combinations of historical and future air temperature and precipitation variations. This demonstrated that the Sary-Tor glacier can always be considered a polythermal glacier since the LIA and into the future, while the Grigoriev ice cap has always been and will always be characterised by a cold structure, despite the ice being significantly warmed in the higher parts of the ice cap when air temperatures increase. The reason for this is that at these higher temperatures, the ice cap will disappear faster than it warms up to a temperate state.

Because the selected ice masses are typical examples of ice bodies in the Inner Tien Shan, and because a detailed sensitivity analysis revealed robust results to changes in the parameters and the magnitude of the geothermal heat flux, it is reasonable to conclude that the results of this study can be generalised for similar type of glaciers and ice caps in the study area. Valley glaciers in the Inner Tien Shan are probably almost all polythermal, while the high-altitude ice caps are expected to be mainly cold. Glaciers in the outer ranges, which are associated to larger amounts of precipitation might be temperate as well. These findings are important as the dynamics of the ice masses can only be understood and modelled precisely if ice temperature is considered correctly in ice flow models. The calibrated parameters of this study can be used in applications with ice flow models for individual ice masses as well as to optimise more general models for large-scale regional simulations.

**Data availability**

Research data is provided through an online public repository, accessible via https://doi.org/10.5281/zenodo.6556313. (Van Tricht et al., 2022). Information and specific details model code will be specified on request by Lander Van Tricht.

**Author contributions**

LVT developed the method, performed the experiments and wrote the manuscript. PH provided the model code, and gave guidance in implementing the research and interpreting the results, assisting during the entire process.

**Competing interests**

The authors have declared that they do not have any competing interests.



**Disclaimer**

/

**Acknowledgements**

The authors would like to thank everyone who contributed to the fieldwork to carry out the ice thickness measurements. We
would also like to specifically thank Benjamin Vanbiervliet for processing the RES data and Oleg Rybak and Rysbek
Satylkanov for assisting and organising the fieldwork. We also thank Ivan Lavrentiev for the GPR data from the Sary-Tor
glacier that was used to distinguish between temperate and cold ice. Finally, we thank Victor Popovnin for providing the mass
balance data of the Sary-Tor glacier.

**Financial support**

Lander Van Tricht holds a PhD fellowship of the Research Foundation – Flanders (FWO-Vlaanderen) and is affiliated with
the Vrije Universiteit Brussel (VUB).

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
