# Peer review of "Thermal regime of the Grigoriev ice cap and the Sary-Tor glacier in the Inner Tien Shan, Kyrgyzstan"

_EGUsphere, 2022_

## Referee Comment (RC2)

**Review to 'Thermal regime of the Grigoriev ice cap and the Sary-Tor glacier in the Inner Tien Shan, Kyrgyzstan'**

**General comments**

The authors apply 3D higher-order thermomechanical ice flow model to describe the thermal regime of Grigoriev ice cap and Sary-Tor glacier both located in the Inner Tien Shan. Historical air temperature and precipitation data, surface elevation and ice thickness measurements as well as the output of a surface energy mass balance model are used to constrain the thermomechanical ice flow model. The modelling results indicate cold conditions for Grigoriev ice cap and a polythermal structure for Sary-Tor glacier. These results agree with previous studies. The thermomechanical ice flow model indicates that the differences in thermal regimes are caused by higher ice velocities (higher horizontal advection rates), larger amounts of insulating snow and higher latent heat release on Sary-Tor. Furthermore, the authors present a sensitivity experiment which highlights that the thermal structure of the ice masses is not constant over time.

These are interesting and important results for a rather poorly studied region where only little knowledge exist about the thermal regime of glaciers. While the purpose of this study is clear and the general approach provides meaningful and relevant results, the clarity of the manuscript should be improved. I have a few general questions, which may point on such ambiguities, and also some general suggestions potentially contributing to the clarity.

In the introduction chapter more context would be helpful. I suggest to provide an overview of previous applications of 3D higher-order thermomechanical ice flow models to determine the thermal regimes of mountain glaciers. Ideally, applications of such models for glaciers with similar conditions (i.e. polythermal glaciers) and related limitations (and solutions) are introduced. This would help to convince the reader that the chosen method is suitable. Have this or similar models been applied to glaciers in the region before? Furthermore, the description of glacier thermal regimes should be improved. I suggest to follow the definitions in the Russian speaking literature based on Shumskii (1964), which is differentiated and therefore suitable for mountain glaciers. Later works by Kotlyakov (1984) and Krenke (1982) may also be worth to be considered here.

Regarding the method applied, I have some questions regarding the chosen boundary conditions. I furthermore suggest to include a quantification of uncertainties related to these choices.

Lower boundary conditions: As visible from figure 11, there is quite a misfit between modelled and measured temperatures at depth. How sensitive are these results to the lower boundary conditions? Have other values be tested? It would be good to expand the discussion regarding this and to visualize results of a related sensitivity experiment.

Upper boundary conditions: I wonder whether warming by latent heat release is correctly represented, especially for Sary Tor uncertainties might be quite high. Regarding the importance of $r_{rem}$ together with $w_f$ (and also $s_{max}$ together with $i_s$), there estimation should be more thoroughly described and related uncertainties should be quantified. $r_{rem}$ is quantified using a simple parametrisation of refreezing (within a mass balance model) which is a function of height and melt only. The model does thus not include any snow and firn characteristics (which are temporally and spatially heterogenous and have a major impact on refreezing), nor calculate internal accumulation. I suggest to expand the section about the mass balance model describing how $r_{rem}$ is estimated. It would be interesting to do simulations for different $r_{rem}$ values in order to quantify related uncertainties. If I understand correctly, potential errors in $r_{rem}$ may be corrected for by tuning $w_f$ to subsurface temperatures on Grigoriev ice cap and furthermore may be averaged out by using

averages over several decades. The uncertainties might thus be reduced if calibration data is available. This could be discussed together with quantification of the sensitivity regarding $r_{rem}$ and comparison to literature values of $r_{rem}$. Potentially, different $r_{rem}$ values could also be discussed in the context of climate change and changing glacier zones (zones by Shumskii, 1964) see my comment above). This would help to interpret the results for Sary Tor, where firn conditions (and refreezing) are likely different (due to different firn conditions related to higher precipitation rates) and for potential other sites, where no calibration data is available.

Besides this, the methodological description would become clearer if somewhat restructured. It is a bit confusing that both chapter 2 and 3 present data descriptions. Chapter 2 seems to mainly refer to historical data. I suggest to either (i) rename chapter 2 to "Study area" and 2.2. to "Historical englacial temperature measurements" or to (ii) move all the data descriptions to chapter 2. Chapter 3 could be a bit better structured by separating background information (which could be moved to the introduction), model description, and data used (or move this to chapter 2) more clearly. The description of the mass balance model deserves more details (parametrisations). When restructuring, the length of subchapters which sometimes seem a bit arbitrarily chosen should be homogenized and the headings could be optimised.

The results chapter presents relevant results and also contains elements of a discussion. Please rename the chapter results and discussion or even better strictly separate results and discussion. The discussion should be a bit more complete. Please also revise whether 4.1. really belongs to the results. It might also be moved to the method chapter, where a section about calibration could be added.

The discussion in chapter 5 could be extended (see also my comments above on the boundary conditions). Furthermore, the uncertainties of the study could be contextualised by referring to previous studies. In section 5.2., I think it would be interesting to discuss a potential firn warning on Grigoriev (as a response to different parameter disturbances) than just saying that the site remains cold. The statement from the abstract 'a detailed analysis concerning the influence of temperature and precipitation changes at the surface reveals that the thermal structure of both ice bodies is not a constant over time, with recent climate change causing more temperate ice in higher areas' could be underlined more with the discussion in 5.2.

In addition to these remarks, I have some more specific and technical comments.

**Specific comments**

L8/9 delete the first sentence of the abstract and complete the second: "An accurate knowledge of the thermal regime of glaciers and ice caps is important to understand their dynamic and response to climate change, and to model their evolution."

L17-19 statement needs to be corrected/rephrased. Not sure, whether the term superimposed ice is correct here. I expect latent heat release being relevant also from other processes than the formation of superimposed ice. Refreezing of melt water above and also below the last summer horizon (internal accumulation) are probably very important processes as described for Sary Tor in Dyurgerov & Mikhalenko (1995) and also described in Kronenberg et al. (2016)and indicated by ice lenses and layers in Grigoriev cores (e.g. Takeuchi et al., 2014; Thompson et al., 1997).

L19 The use of the term 'ice surface temperature' might be confusing, see also later comment.

L30-32 Please be a bit more specific here. I suggest to add one or two sentences + references why such knowledge is needed and how the thermal regime affects the response of ice bodies to climate change.

L38&L41 As Abramov glacier is located in the Pamir Alay and not in the Tien Shan, I am not sure whether it should be mentioned here. If so, I suggest to write something like 'Abramov glacier located in the nearby Pamir Alay and rather refer to the subsurface temperature measurements directly (Kislov et al., 1977), which indicated cold temperatures in the ablation area and temperate conditions in the accumulation area. Please also note, that in the uppermost meters, seasonal cooling is quite substantial on Abramov glacier, but all the cold content in the accumulation area is 'consumed' by refreezing of melt water leading to temperate conditions (cf. Kronenberg et al., 2022).

L51-54 Please verify statements and cited references here. At least Kronenberg et al. (2021) does not state this. In the infiltration-congelation zone, the refreezing of melt water is limited by the amount of available pore space and not by temperatures which are rather cold (than temperate) here (Shumskii, 1964).

L54-56 statement unclear. Do you refer to the infiltration zone as defined in Kotlyakov, (1984); Krenke (1982)? What is thawed melt-water? Depending on the site, infiltration-congelation zone is the lower-most zone (cf. Shumskii, 1964).

L80 calving → dry calving

L86 delete 'on the other hand'

L86 what is small? Better provide area in km$^2$

L106-114 this rather belongs to the introduction than to the data section. Also, it seems that some statements are repeated from above

L114 reference needed

L120 warm →temperate

L135 last → deep

L135 on the top →near to the summit

L142 delete last statement

L151 what kind of mountain glaciers? What climatic environment? Maybe also provide this information in the introduction…

L151-153 was there a study using the same model and providing a detailed description? Would be good to state here, which study this was and write something like: "In the following the model is briefly described. Please refer to xx and references therein for a more detailed description. "Despite this, make sure that the reader knows the basis of your choices such as chosen parameters etc.

L164 why 3? Reference

L164 and also elsewhere always write (eq. 2) and not just (2) when referring to equations

L167 I am not sure whether this should be written here or whether there might a better place to describe the spatial resolution. Might also be good to recall what the average layer thickness is (for both sites)

L175 provide reference

L202 provide typical values of m for typical conditions…

L235 (Barandun et al., 2015) is in the Pamir Alay. I don't know whether this is relevant here.

L249-251 verify statement/definition of the zone (see also previous comments)

L256 Here and probably also later, it might be clarifying to write 'surface layer' instead of 'surface' e.g. "…meltwater refreezing and its warming effect on the glacier surface layer"

L266 Rather provide an example from a glacier or better introduce this, otherwise the reader is very surprised that you start to talk about permafrost here..)

L276-280 Make sure, that this paragraph really states what you would like to, what is not fully clear as it is written now).

L285 'for all the effects described above' is unclear

L290 why sinusoidal and not monthly lapse rates provided by (Aizen et al., 1995)

L293 refer to section describing mass balance model (in more detail than it is currently done).

L297 It would be good to justify this choice, as most precipitation is falling during summer months cf.(Dyurgerov et al., 1994; Kronenberg et al., 2016)

L342-343: Why does the water refreeze at the end of the mass balance year? What are the implications of this assumption and of the assumption of the chosen threshold of 60 %? How does this estimation compare to more sophisticated simulations of refreezing? And what is about refreezing below the last summer surface?

L347-348: Can this choice be underlined by in situ measurements? (See also comment above. To my knowledge, most precipitation is falling during summer months and I would therefore assume substantial snow mass being accumulated after May.)

L375: Can you please quantify the uncertainties of this assumption? Why not use a DEM based on optical data?

Section 3.5.4: I suggest to delete this section and to add the relevant information to section where you describe the outlines.

L421 15% larger than what?

L422 ~30% more than what?

L423 The values given for Sary Tor here are different from the ones given above. Is this due to your correction based on melt? I don't fully understand this correction and it is difficult to get, why the values are presented once more.

L426 refer to a map which shows this eastern tributary

L438-443 These are not results. Also, Grigoriev ice cap is strongly wind-exposed and wind erosion may also play a major role here.

L450/451 It is not very clear here, where these values are coming from. Are they optimised to obtain a better fit between the measured and modelled glacier extent for 1990. (I assume so from the statement in L397). This somehow implies that the mass balance model does not provide suitable mass balance estimates. Does this not also question the mass balance model estimates of the refreezing?

L466 please provide units. And maybe recall, what $w_f$ stands for (firn warming parameter).

L466ff this is rather a discussion than results.

L467 I doubt that this is a valid argument here. Refreezing won't take part at depth, but only in the firn. Furthermore, refreezing may be reduced due impermeable ice layers cf. (Machguth et al., 2016)

L472 please provide units.

L482 I think this is rather related to the lack of firn pore space than due to a thin snow pack. The firn is likely thinner and with higher densities in the lower accumulation area.

L486 Summer precipitation is also mostly snow (cf. e.g. Dyurgerov et al., 1994, Kronenberg 2016).

L488 surface →surface conditions (2x)

L493 ensures a larger cooling of the surface → allows for a larger surface cooling

L512 'The enhancement factor is set at 3 for the Grigoriev ice cap and at 4 for the Sary-Tor glacier, which appears to be the optimal value to match with the reconstructed ice thickness of 1990.' Is not really a result. I suggest to rather provide this information within the method chapter adding a section where you describe the calibration.

L515-522 Please shorten this paragraph saying the modelled and reconstructed topographies match well and discrepancies are mainly occurring in areas where no GPR measurements exist.

L525 How do your measurements of the ice thickness compare to the deep core which was drilled relatively near to the summit (Takeuchi et al., 2014)

L527 Please provide a refence for the statement about the complexity of modelling ice thicknesses at ice divides

L529 Please provide a reference for the enhancement factor of 1 corresponding to -5°C (rather write °C than 'degrees').

L542 Please refer to the map in fig 8c it here.

L561 Please add here, when these observations were performed.

L572 Please use full variable names and give short forms in brackets.

L573: Should an improved transport of geothermal heat not rather lead to a warming?

L576 ensures →shows?

L579-580 Please rephrase 'This ensures a greater stiffness which causes the ice intrinsically slower to deform.'

L584/585 Not clear, why temperate ice causes the infiltration of water. Water can either infiltrate into ice through crevasses, moulins or infiltrate into the firn unless an impermeable ice layer is

reached within the firn (Machguth et al., 2016). Such infiltration occurs in cold and temperate firn (Shumskii, 1964).

L587 'This is the case for the entire surface of the Grigoriev ice cap'. This is not correct. Water percolation of into the subsurface also occurs on Grigoriev as e.g. visible from ice lenses within the firn in different cores from (Takeuchi et al., 2014; Thompson et al., 1997)

L613/614 This statement needs a reference. And, also, it is not fully clear why this is written here. I suggest to rephrase the paragraph so that the argumentation becomes clearer.

L627 Where do the mean climatic conditions for 1920-1950 come from? Does the Kumtor time series not start in the 1930s?

L630 Could the mismatch in fig. 11 between simulations and measurements not also be (partly) related to a wrong lower boundary condition and/or the used temperature gradient? Would be interesting to visualize of the corresponding sensitivity experiment here.

L675 the way, how $w_f$ and especially $i_s$ are calculated also has uncertainties. This should be discussed here, as the overall temperature regime seems to be very sensitive to $i_s$

L705 From Figure 12 it seems, that warming temperatures cause an increase of the percentage of temperate ice on Grigoriev. So, the statement that it remains cold should be corrected. Also, do your simulation show a warming happening with warmer temperatures. Even if temperatures stay below the melting point, a warming may be happening – would be interesting to briefly discuss this here and maybe put into context with observed and modelled firn warning in the alps (Hoelzle et al., 2011; Mattea et al., 2021; Vincent et al., 2020)

L717 not only the formation of superimposed ice (which happens at the edge of the accumulation area) releases heat, but also refreezing of melt water within the firn is a relevant warming process.

L718 Please change 'This has already been observed at the Abramov glacier in the southwestern Tien Shan (Kronenberg et al., 2020).to 'Evidence of a precipitation increase has been observed on Abramov glacier in the Pamir Alay (Kronenberg et al., 2021)'

L735 'reduced formation of refrozen meltwater at the end of summer' → 'reduced refreezing of melt water' (Refreezing is an ongoing process, not restricted to the end of summer. Especially on temperate sites, it happens at the beginning of the melt season (cf. Kronenberg et al., 2022)

L737 This may be the case in the model applied here, which does not account for firn. But is not true in general. Melt water is known to infiltrate to the subsurface also under cold conditions (Shumskii, 1964).Please clarify here.

L766 Sary-Tor →Grigoriev

L870 2020 →2021

**Technical corrections**

References: Please add references to statements where they are missing and make sure that all references are given following the journal's guidelines. Sometimes, the year is provided in brackets, sometimes not.

The language of manuscript could be improved I recommend a proofreading by a native speaker.

General comment on the tables: Please homogenize the table layout. I suggest to use a typical layout without coloured lines, shaded areas etc.

General comment on figures: Please add panel letters to all the figures which show more than one plot/map and refer to those letters in the captions. Use larger fonts in legends and labels. Plot legends aside and not on the top of maps/plots and show them in a reasonable size (they are small and difficult to read). Place labels on maps more carefully so that the they are better visible. Complete information in caption (year of glacier outlines is usually missing, say that the legend is for all the subplots etc.).

Table 2: provide units also in the table (and not only in the caption). Does 1986-1987 refer to the mass balance year starting from 1.10.1986 and ending on the 30.9.1987? If so rather write 1986/87. And replace 'Average 1963-1989' with 'average 1963/64-1988/89'. Provide reference in a sperate column. Also, it seems strange that the glacier wide value is given in the centre. Either provide it first or last. And 'at the front' is very unspecific. Due you refer to a point measurement here or to some averaged value? Please show on a map where this is and specify, whether always the location is used.

Figure 1: add legend with information about of background topography

Figure 3: Zoom in to Sary Tor. The glacier is quite small and you loose space by showing its surroundings. Visualize core locations on Grigoriev map.

Figure 5b display x and y axes with the same length.

Figure 6: Provide x/y axes for all panels or say in the captions that they stand for all. Improve legend!

Figure 8: If D in panel d corresponds to D in panel c, 'A' should be replaced with 'C' in panel c.

Figure 9: cross-reference in caption wrong (should be figure 8 or 7). Add letters: A,B,C,D

Figure 10 Clarify the colour bar. Is the darkest red colour tone in the left panel referring to temperatures at 0°C. What does the second class stand for? Values between -0.5 and -1°C? Plot the simulated boundary between temperate and cold ice in the right panel. And show both panels at equal spatial resolution

Figure 11: was the purple profile simulated by a stepwise temperature increase or was the change applied once?

Figure 12: homogenize font sizes of axes labels

**References**

Aizen, V. B., Aizen, E. M., & Melack, J. M. (1995). Climate, Snow Cover, Glaciers, and Runoff in the Tien Shan, Central Asia. *Water Ressources Bulletin*, *31*(6), 1113–1129. https://doi.org/10.1111/j.1752-1688.1995.tb03426.x

Barandun, M., Huss, M., Sold, L., Farinotti, D., Azisov, E., Salzmann, N., Usubaliev, R., Merkushkin, A., Hoelzle, M., A.Merkushkin, & Hoelzle, M. (2015). Re-analysis of seasonal mass balance at Abramov glacier 1968-2014. *Journal of Glaciology*, *61*(230), 1103–1117. https://doi.org/10.3189/2015JoG14J239

Dyurgerov, M. B., & Mikhalenko, V. N. (1995). *Glaciation of Tien Shan [in Russian]* (M. B. Dyurgerov & V. N. Mikhalenko, Eds.). VINITI.

Dyurgerov, M. B., Mikhalenko, V. N., Kunakhovitch, M. G., Ushurtsev, S. N., Chaohai, L., & Zichu, X. (1994). On the cause of glacier mass balance variations in the Tien Shan mountains. *GeoJournal*, *33*(2--3), 311–317.

Hoelzle, M., Darms, G., Lüthi, M. P., & Suter, S. (2011). Evidence of accelerated englacial warming in the Monte Rosa area, Switzerland/Italy. *The Cryosphere*, *5*, 231–243. https://doi.org/10.5194/tc-5-231-2011

Kislov, B. v, Nozdrukhin, V. K., & Pertziger, F. I. (1977). Temperature regime of the active layer of Abramov Glacier [in Russian]. *Materialy Glatsiologicheskih Issledovanii (Data of Glaciological Studies)*, *30*, 199–204.

Kotlyakov, V. M. (1984). *Glaciological Dictionary [in Russian]* (UDK 55.132, pp. 1–564). Gidrometeoizdat.

Krenke, A. N. (1982). *Mass exchange in glacier systems in the USSR [in Russian]*. Gidrometeoizdat.

Kronenberg, M., Barandun, M., Hoelzle, M., Huss, M., Farinotti, D., Azisov, E., Usubaliev, R., Gafurov, A., Petrakov, D., & Kääb, A. (2016). Mass-balance reconstruction for Glacier No. 354, Tien Shan, from 2003 to 2014. *Annals of Glaciology*, *57*(71), 92–102. https://doi.org/10.3189/2016AoG71A032

Kronenberg, M., Machguth, H., Eichler, A., Schwikowski, M., & Hoelzle, M. (2021). Comparison of historical and recent accumulation rates on Abramov Glacier, Pamir Alay. *Journal of Glaciology*, *67*(262), 253–268. https://doi.org/10.1017/jog.2020.103

Kronenberg, M., Machguth, H., Pelt, W. van, Fiddes, J., Hoelzle, M., & Pertziger, F. (2022). Long-term mass balance and firn modelling for Abramov glacier, Pamir Alay. *The Cryosphere Discussions*, *2021–380*, 1–33. https://doi.org/10.5194/tc-2021-380

Machguth, H., MacFerrin, M., van As, D., Box, J. E., Charalampidis, C., Colgan, W., Fausto, R. S., Meijer, H. A. J., Mosley-Thompson, E., & van de Wal, R. S. W. (2016). Greenland meltwater storage in firn limited by near-surface ice formation. *Nature Climate Change*, *6*, 390–393. https://doi.org/10.1038/nclimate2899

Mattea, E., Machguth, H., Kronenberg, M., van Pelt, W., Bassi, M., & Hoelzle, M. (2021). Firn changes at Colle Gnifetti revealed with a high-resolution process-based physical model approach. *The Cryosphere*, *15*, 3181–3205. https://doi.org/https://doi.org/10.5194/tc-15-3181-2021

Shumskii, P. A. (1964). *Principles of structural glaciology*. Dover Publications Inc.

Takeuchi, N., Fujita, K., Aizen, V. B., Narama, C., Yokoyama, Y., Okamoto, S., Naoki, K., & Kubota, J. (2014). The disappearance of glaciers in the Tien Shan Mountains in Central Asia at the end of Pleistocene. *Quaternary Science Reviews*, *103*, 26–33. https://doi.org/10.1016/j.quascirev.2014.09.006

Thompson, L. G., Mikhalenko, V. N., Mosley-Thompson, E., Dyurgerov, M. B., Lin, P. N., Moskalevsky, M., Davis, M. E., Arkhipov, S., & Dai, J. (1997). Ice core records of recent climatic variability: Gregoriev and It-Tish ice caps, in central Tien Shan, Central Asia. *Materialy Glatsiologicheskih Issledovanii (Data of Glaciological Studies)*, *81*, 100–109.

Vincent, C., Gilbert, A., Jourdain, B., Piard, L., Ginot, P., Mikhalenko, V., Possenti, P., le Meur, E., Laarman, O., & Six, D. (2020). Strong changes in englacial temperatures despite insignificant changes in ice thickness at Dôme du Goûter glacier (Mont Blanc area). *The Cryosphere*, *14*, 925–934. https://doi.org/10.5194/tc-14-925-2020

---

## Author Comment (AC1)

In this document, we respond to the comments of reviewer 1 one by one. Whenever some entirely new text has been added to the manuscript, it has been added in italics and in red.

**Reviewer 1**

This manuscript presents a detailed exploration of the thermal regime of the Grigoriev ice cap and the Sary-Tor glacier, which are located in the Inner Tien Shan in Kyrgyzstan. Using a wide range of observations, such as observed temperature profiles and GPR profiles, model parameters are tuned and subsequently evaluated against observations. The study finds that the Sary-Tor can always be considered a polythermal glacier unlike the Grigoriev ice cap which can always be considered as a cold structure. It is also suggested that the found parameters can be generalized to similar type of glaciers in the region. The author(s) did a great job reviewing and referencing previous work. Overall this was found to be a very extensive and thorough study.

We would like to thank the reviewer for the useful review which helped us to improve the quality of the manuscript.

**Specific comments**

[RC1.1] Geothermal heat. The Geothermal heat flow parameterization used seems too casual. The study is using the same constant value for the geothermal heatflux for both the valley glacier and the ice cap, however, it has been showed that geothermal heatflux is focused in valleys like Sary-Tor and diminished on ridges, like Grigoriev Ice Cap. It is possible that Grigoriev Ice Cap has double the heat flow as Sary-Tor. See the papers by Colgan et al. 2020 (doi:10.1029/2020JF005598), section on topographic correction and Van der Veen et al. 2007 (doi:10.1029/2007GL030046). The chosen value for the geothermal heat boundary condition of 50mW/m2 seems quite high given the altitude of the glaciers. Another study by Zhong et al. 2013 from Journal of glaciology doi:10.3189/2013JoG12J202, studying the East Rongbuk Glacier, have derived a lower geothermal heatflux value of approximately 19mW/m^2.

We would like to thank the reviewer for these relevant suggestions. We kept the average heat flux of 50 mW m$^{-2}$ as different previous research showed this value to be the average geothermal heat flux in our study region. We added two more references to previous studies on this subject (Duchkov et al., 2001; Vermeesch et al., 2004). The East Rongbuk Glacier is located in the Himalaya, and it is also situated about 3000 meters higher which may explain the difference between the geothermal heat flux used. Furthermore, the Tien Shan is assumed to be associated with crustal thickening and thus is expected to display an increased surface heat flow (Vermeesh et al., 2004).

Two references were added:

*Duchkov, A.D., Yu.G. Shvartsman, and L.S. Sokolova, Deep heat flow in Tien Shan: advances and drawbacks, Geologiya i Geofizika (Russian Geology and Geophysics), 42, 10, 1516–1531(1436–1452), 2001.*

*Vermeesch, P., Poort, J., Duchkov, A., Klerkx, J. and De Batist, M.: Lake Issyk-Kul (Tien Shan): Unusually low heat flow in an active intermontane basin. GEOLOGIYA I GEOFIZIKA, 45(5), 616–625, 2004*

Following your suggestion, we implemented a topographical correction following Colgan et al. (2021). As such, we account for the influence of topographic relief on the geothermal heat flux within the study region.

*Colgan W., MacGregor J.A., Mankoff K.D., Haagenson R., Rajaram H., Martos Y.M., Morlighem M., Fahnestock M.A. and Kjeldsen K.K.: Topographic correction of geothermal heat flux in Greenland and Antarctica, Journal of Geophysical Research. Earth Surface, 126, https://www.doi.org/10.1029/2020JF005598, 2021*

In lines 446-461, we added:

*To account for the influence of topography on the geothermal heat flux, we apply the empirically determined topographic correction procedure described in Colgan et al. (2021). This concerns a high-pass filter with a dimensionless correction factor applied to the average geothermal heat flux, which makes the geothermal heat flux spatially variable depending on the local elevation. Using the correction factor, the geothermal heat flux is magnified in incised valleys such as for the Sary-Tor glacier and attenuated on ridges such as for the Grigoriev ice cap. More specific, in the correction procedure, the average geothermal heat flux is perturbed by an anomaly $\left(\frac{\Delta g_{hf}}{\overline{g_{hf}}}\right)_{ij}$ to obtain the local geothermal heatflux ($g_{hf,ij}$).*

$$g_{hf,ij} = \overline{g_{hf}}\left(1 + \frac{\Delta g_{hf}}{\overline{g_{hf}}}\right)_{ij} \tag{14}$$

*The anomaly is estimated as a function of local relief using*

$$\left(\frac{\Delta g_{hf}}{\overline{g_{hf}}}\right)_{ij} = \frac{1}{\alpha}(\bar{h}_{ij} - h_{ij}) \tag{15}$$

*with $\alpha$ an empirically determined characteristic height (950 m) and $\bar{h}_{ij}$ the mean elevation averaged within a moving window of 10 x10 km centred over location ij.*

Applying this topographic correction, the obtained geothermal heat flux on the summit of the Grigoriev ice cap is substantially lower (29 mWm$^{-2}$), which is more in line with observation of Arkhipov et al. (2004) that the ice temperature at the summit did practically not depend on the vertical coordinate at depths from 10-45 m (~lower geothermal heat flux).
* * *
**[RC1.2]** Line 650-655: There is something unclear in this section. First it is described that average ice temperatures are most sensitive to changes in geothermal heat, but the paragraph is ended with 'geothermal heatflux do not have any major influence'. I assume the latter statement is the conclusion the author draws from the sensitivity analysis, and the former just means that relative to the other tested parameters geothermal heat influence the mean temperature the most. Maybe the section can be written slightly different?

We agree with the reviewer and changed this paragraph in lines 1261-1264:

*"For both ice bodies, the average ice temperature changes only slightly when the parameters and the average geothermal heat flux are altered within the predefined ranges (Table 4). Hence, based on the sensitivity analysis, it can be concluded that for the selected ice masses, parameter uncertainty and the magnitude of the geothermal heat flux do not have a major influence on the obtained thermal regime, making the results of this study robust."*

**[RC1.3]** Temperature calculations. As far as I can tell the ice flow model is a 'cold ice' model like most ice flow models. Modelling the flow of temperate and polythermal ice is different to modelling cold ice since it is essential to know the spatial distribution of the water content in the ice (See Dynamics of Ice sheets and Glaciers by Ralf Greve and Heinz Blatter). According to a study by Andy Aschwanden (doi:10.3189/2012JoG11J088) 'cold-ice' models are not energy-conserving when used with temperate ice since they cannot account for the part of the internal energy which comes from latent heat of liquid water. This could be included as a discussion point. Technically both glaciers could be considered polythermal, following the IACS glacier terminology Cogley 2011.

We agree with the reviewer, and we added a section in the methodology section describing that the applied model is a cold ice model in lines 337-341:

*"The applied HO model is a cold ice model which implies that it is not energy conserving when temperate ice is present, as it does not account for the part of internal energy coming from the latent heat of liquid water (Aschwanden et al., 2012, 2017). Hence, it does not take into account the presence of water in the ice. Nevertheless, the tuning of the enhancement factor (m) is an implicit way to include softening effects due to factors such as water content and impurity content (Huybrechts et al., 1991)."*

We are aware that technically, if the temperate ice of the Grigoriev ice cap is not only present at the contact surface between the ice and the bedrock, we can technically classify this ice cap as being polythermal. However, since the surface area and volume for which temperate ice was found is very limited (< 0.01% under standard conditions, and only the bottom layer in contact with the bedrock), the ice cap is predominantly cold, and we decided to preserve the terminology.

**[RC1.4]** Line 309-311: The explained effect is also described in Hooke 1976 J. glaciology, study of Barnes ice cap. It might be good to cite that paper.

Thank you for the suggestion. We added a reference to Hooke (1976) in line 511.

**General questions and comments on unit, equations and notation:**

**[RC1.5]** There are many parameters and m appears several times it might be nice to include an annotation table for clarity? This could also help to clearly define units of every variable. Sometimes variables in text does not have units e.g. the melting point depression factor, gamma in line 198.

We agree with the reviewer and added a table with all variables, constants, and their symbols, values and units (Table 1).

**[RC1.6]** How is the internal heat production P from equation (9) calculated?

P is calculated from strain heating. We added this in the text on line 395: *"The internal heat production (P) is calculated from strain heating (Huybrechts, 1996)."*

**[RC1.7]** Where does equation (15) come from?

We added a sentence to make this clearer in lines 461-462:

*"Then, as boundary condition at the bed, the temperature gradient is defined as the sum of the local geothermal heat flux and the heat due to basal sliding (Huybrechts and Oerlemans, 1988)."*

**[RC1.8]** Equation (16) The last term of the surface temperature ($T_s$) parameterization for H<ELA does not seem to have units of degree if ms has units of m w.e. pr yr? units don't seem to match with $T_s$ having units of degree Celsius.

Thank you for this remark. The '10' value in the denominator of the fraction has unit metres of ice equivalent. We added the units to $m_s$ and 10 in this equation to make it clear for the reader.

**[RC1.9]** The introduction mentioned that Grigoriev Ice Cap is losing mass by calving, but this is not mentioned in the mass balance model section. Does the 'm_s at the front' in table 2 refer to mass loss from calving? What do the tuning parameters c0 and c1 represent, I don't think they are explained fully in the text? It seems like one would have to read Van Tricht et al. 2021b to understand it the section does not stand completely on its own. It might be helpful to explain a bit further?

The mass loss by calving for the Grigoriev ice cap (at the north side) is situated at the highest elevations (>4500 m a.s.l.). It is not included in the mass balance model. To make this clearer, we added in lines 771-772:

*"Concerning the northern boundary of the Grigoriev ice cap, when ice flows beyond the predefined calving front, it is automatically removed."*

Concerning more explanation about the mass balance model, we added in lines 619-621:

*"This simple model is based on incoming solar radiation, temperature and precipitation and contains two additional tuning parameters ($c_0$ and $c_1$) representing the sum of the longwave radiation balance and the turbulent sensible heat exchange."*

And we repeated in line 627:

*"($c_0$ and $c_1$)"*

We also added the parameters in the annotation table (Table 1).

**[RC1.10]** Line 396: could we get more information on how the mass balance correction works? I assume it is not spatially constant correction in order reach the 1990 geometry but the text says 'constant correction'?

We use a constant mass balance correction which means that we add/subtract over the entire glacier/ice cap a constant mass balance value. To clarify this, we added *"over the entire ice mass"* in line 782.

**[RC1.11]** Line 415-418: No error bars are given on the area and volume estimates?

Thank you for this suggestion. We added error bars following the same approach as in Van Tricht et al. (2021a) to estimate the uncertainty of the reconstructions.

**[RC1.12]** Figure 5. Very nice sensitivity plot – as I understand from the figure these values are constant in space and time? It might be nice to make that clear in the text, it also means there is potential for further work for examining the spatial variability of the parameters i_s and w_f.

Thank you. We added a sentence in lines 980:

*"In the applied model, the calibrated values of $w_f$ and $i_s$ are kept constant in space and time."*

**[RC1.13]** Line 512: why is the enhancement factor chosen to be different (3 and 4 respectively) for 2 glaciers which are stated to be similar?

The enhancement factor is calibrated to match the modelled thickness optimally with the ice thickness reconstructions. The enhancement factor indirectly takes into account softening effects actors such as water content. This is why the Sary-Tor glacier and the Grigoriev ice cap, although located close to each other, can have a slightly different value for m.

**[RC1.14]** Would it be possible to include an evaluation of modelled velocities against observed velocities e.g. using the ITS_LIVE product? If it was showed the velocities were reasonably well captured this would also strengthen the confidence in the calculated temperatures. approximately line 540

We compared the modelled velocities with observations from mass balance stake displacements and we found a close correspondence. We added this in the text in lines 1090-1091:

*"Comparison between modelled velocities and velocities derived from mass balance stake displacements at the Sary-Tor glacier shows a close correspondence (RMSE = 1.5 m yr$^{-1}$)."*

**[RC1.15]** Line 569: It is written that: 'the difference between the thermal regime of the two ice masses can be attributed to several factors' – this also includes geothermal heat focused in valleys like Sary-Tor and diminished on ridges like Grigoriev ice cap (see Colgan 2021, JGR), topographic corrections.

This is correct. We added in line 1107: *"the geothermal heat flux corrected for topographic relief"*

**[RC1.16]** It would be helpful if you include a description of how it possible to distinguish between temperate and cold ice from the radargram.

Thank you for the suggestion. We added a short description in the caption in lines 1163-1165:

*"The figure clearly shows two sections of cold (upper part) and temperate (lower part) ice. The latter is characterised by the presence of small hyperbolic diffraction features due to the presence of water in the ice, typical for temperate (shown here as warm) ice."*

**[RC1.17]** Financial support section: Kumtor mining company has been funding glaciology in this valley. Maybe include a clear statement that no industry funding was used or who did pay for the logistics of the field work?

*We added a sentence in lines 1475-1476: "Local logistics were organised and funded by the Tien Shan High Mountain Research Center and the Kumtor mining company."*

**(Minor) Technical corrections and suggestions**

**[RC1.18]** Line 38: using GPR abbreviation but it is not explained until line 64-65.

Thank you for this remark. We explained the abbreviation in line 51.

**[RC1.19]** Line 117: 'down to a depth' instead of 'up to a depth of 102 m'?

Done.

**[RC1.20]** Line 135: 'down to bedrock' instead of 'up to bedrock'

We followed your suggestion and replaced 'up to the bedrock' into 'down to the bedrock'.

**[RC1.21]** Line 136: 'depth' missing p

We added a 'p'

**[RC1.22]** Line 164 +166: include Eq. when referring to equation (2) and (4) to be consistent with the other times you refer to an equation.

Done.

**[RC1.23]** Line 289: 'assumed not to be' instead of 'to be not'

Done.

**[RC1.24]** Line 440: 'On' instead of 'At' in sentence 'At the other hand…'

Modified.

**[RC1.25]** Line 511: the number of decimal places vary

We added a second decimal for the RMSE of the Grigoriev ice cap.

**[RC1.26]** Line: 622: write '1960' instead of 'the sixties'

Done.

**[RC1.27]** Line 669: missing space between table and 1.

Modified.

**[RC1.28]** If possible it would be nice to see the location of the observed temperature profiles on a map.

We added the location of the borehole of 2007.

**[RC1.29]** Equation (7) should be A(T_pmp) instead of A(T)

Replaced.

**[RC1.30]** Figure 1: The light blue colour indicating the glaciers are not very clear, it is very similar to the blue showing lower elevations of the background map.

We changed the light blue colour to a white colour to create a greater contrast between the lower elevations and the glaciers/ice caps.

**[RC1.31]** Figure 5 right figure: suggest removing 'at 10-15m depth' from the x-axis label

Done.

**[RC1.32]** Figure 8:
* subfigure c has wrong label of the flowline, should be C and D instead of A and D to fit subfigure d.
* figure text missing f in flowlines in line starting with (b,d) Horizontal ice velocities along the flowlines
* plot elevation on y-axis of subplots (b,d) to be consistent with the other similar figures

- Done.
- Adjusted
- We added the elevation on the y-axis

**[RC1.33]** Table 3
* Initial values for Sary-Tor glacier have %value on top and temperature below to be consistent
* Suggestion: include the actual parameter values in the table not just the relative change

- We replaced the % value and the temperature to be consistent with the layout for the Grigoriev ice cap.
- We agree and now include the actual parameter values in the table (Table 4).

---

## Author Comment (AC2)

In this document, we respond to the comments of reviewer 2 one by one. Whenever some entirely new text has been added to the manuscript, it has been added in italics and in red.

**Reviewer 2**

The authors apply 3D higher-order thermomechanical ice flow model to describe the thermal regime of Grigoriev ice cap and Sary-Tor glacier both located in the Inner Tien Shan. Historical air temperature and precipitation data, surface elevation and ice thickness measurements as well as the output of a surface energy mass balance model are used to constrain the thermomechanical ice flow model. The modelling results indicate cold conditions for Grigoriev ice cap and a polythermal structure for Sary-Tor glacier. These results agree with previous studies. The thermomechanical ice flow model indicates that the differences in thermal regimes are caused by higher ice velocities (higher horizontal advection rates), larger amounts of insulating snow and higher latent heat release on Sary-Tor. Furthermore, the authors present a sensitivity experiment which highlights that the thermal structure of the ice masses is not constant over time.

These are interesting and important results for a rather poorly studied region where only little knowledge exist about the thermal regime of glaciers. While the purpose of this study is clear and the general approach provides meaningful and relevant results, the clarity of the manuscript should be improved. I have a few general questions, which may point on such ambiguities, and also some general suggestions potentially contributing to the clarity.

We would like to thank the reviewer Marlene Kronenberg for taking the time to provide a useful, comprehensive, and detailed review of our paper. We have addressed all comments below and updated the manuscript accordingly where needed. We believe this has strongly improved the clarity and quality of the research.

**General comments:**

**[RC2.GC1]** In the introduction chapter more context would be helpful. I suggest to provide an overview of previous applications of 3D higher-order thermomechanical ice flow models to determine the thermal regimes of mountain glaciers. Ideally, applications of such models for glaciers with similar conditions (i.e. polythermal glaciers) and related limitations (and solutions) are introduced. This would help to convince the reader that the chosen method is suitable. Have this or similar models been applied to glaciers in the region before? Furthermore, the description of glacier thermal regimes should be improved. I suggest to follow the definitions in the Russian speaking literature based on Shumskii (1964), which is differentiated and therefore suitable for mountain glaciers. Later works by Kotlyakov (1984) and Krenke (1982) may also be worth to be considered here.

To date, few 3D thermomechanical model studies (higher-order or full stokes) have been performed to study the thermal regime of glaciers (and ice caps). None of the existing 3D model studies were conducted in the Tien Shan. We added several sentences and references in the introduction section about previous studies that modelled the thermal regime of glaciers and ice caps using 3D models in lines 128-133:

*"To date, most of the 3D glacier and ice cap model studies have been performed assuming an isothermal state of the ice mass (e.g. Jouvet et al., 2011; Zekollari et al., 2014). Only a few studies have been performed applying a 3D thermomechanical ice flow model including variations in englacial temperatures for glaciers (Zwinger et al., 2007; Zwinger and Moore, 2009; Zhao et al., 2014; Rowan et al., 2015; Li et al., 2017; Gilbert et al., 2017, 2020), and ice caps (Flowers et al., 2007; Schäfer et al., 2015; Zekollari et al., 2017)."*

Thank you for the suggestion of the Russian studies. We added several references to Shumskiy (1955) and modified the infiltration congelation zone into the infiltration recrystallisation zone.
* * *
**[RC2.GC2]** Regarding the method applied, I have some questions regarding the chosen boundary conditions. I furthermore suggest to include a quantification of uncertainties related to these choices. Lower boundary conditions: As visible from figure 11, there is quite a misfit between modelled and measured temperatures at depth. How sensitive are these results to the lower boundary conditions? Have other values be tested? It would be good to expand the discussion regarding this and to visualize results of a related sensitivity experiment.

As was suggested by Reviewer 1, we included a topographic correction in the determination of the geothermal heat flux (see RC1.1). Following your suggestion, we included the modelled temperature profiles obtained by using 50% and 200% of the initial geothermal heat flux (as was done in the sensitivity experiment) (see Figure 11).
* * *
**[RC2.GC3]** Upper boundary conditions: I wonder whether warming by latent heat release is correctly represented, especially for Sary Tor uncertainties might be quite high. Regarding the importance of rrem together with wf (and also smax together with is), there estimation should be more thoroughly described and related uncertainties should be quantified. rrem is quantified using a simple parametrisation of refreezing (within a mass balance model) which is a function of height and melt only. The model does thus not include any snow and firn characteristics (which are temporally and spatially heterogenous and have a major impact on refreezing), nor calculate internal accumulation. I suggest to expand the section about the mass balance model describing how rrem is estimated. It would be interesting to do simulations for different rrem values in order to quantify related uncertainties. If I understand correctly, potential errors in rrem may be corrected for by tuning wf to subsurface temperatures on Grigoriev ice cap and furthermore may be averaged out by using averages over several decades. The uncertainties might thus be reduced if calibration data is available.

The $r_{rem}$ value is not a constant. It is varies between 0 and $P_{max}$ (= 0.6) * snowdepth. We kept the 0.6 (60%) as a constant as previous research (Reeh, 1991; Braithwaite et al., 1994) showed this to be the amount of average pore spaces in snow. As the reviewer correctly states in RC2.33,34,35,49, $s_{max}$ is not reached at the end of winter but at the end of spring which we corrected in the manuscript in lines 565 and 954. Furthermore, we added in lines 656-657

"*percolate into the snow cover and to refreeze*".

In our study, we used several borehole temperature measurements from different periods (1962 – 2007) on the Grigoriev ice cap to calibrate $w_f$ and $i_s$. Hence, we assume that indeed errors associated with $r_{rem}$ and $s_{max}$ are smoothed out, especially over longer time scales.

We added a more elaborate discussion about the uncertainty related to $P_{max}$ and $s_{max}$ (see RC2.66).

We also added a longer description in the mass balance section describing the estimation of $r_{rem}$.

**[RC2.GC4]** This could be discussed together with quantification of the sensitivity regarding rrem and comparison to literature values of rrem. Potentially, different rrem values could also be discussed in the context of climate change and changing glacier zones (zones by Shumskii, 1964) see my comment above). This would help to interpret the results for Sary Tor, where firn conditions (and refreezing) are likely different (due to different firn conditions related to higher precipitation rates) and for potential other sites, where no calibration data is available. Besides this, the methodological description would become clearer if somewhat restructured. It is a bit confusing that both chapter 2 and 3 present data descriptions. Chapter 2 seems to mainly refer to historical data. I suggest to either (i) rename chapter 2 to "Study area" and 2.2. to "Historical englacial temperature measurements" or to (ii) move all the data descriptions to chapter 2. Chapter 3 could be a bit better structured by separating background information (which could be moved to the introduction), model description, and data used (or move this to chapter 2) more clearly. The description of the mass balance model deserves more details (parametrisations). When restructuring, the length of subchapters which sometimes seem a bit arbitrarily chosen should be homogenized and the headings could be optimised.

Done. See RC2.GC3 and RC2.66.

- We renamed section 2 "Study area" and 2.2 "Historical englacial temperature measurements".
- We renamed section 2 "Methods and data" as it also incorporates description of the data used in the study.
- We moved part of the text on calibration of the mass balance model to section 3.4)
- Besides, we have expanded the mass balance section, and we added more information about the refreezing in lines 620-622 (see RC2.GC3).
  *"This simple model is based on incoming solar radiation, temperature and precipitation and contains two additional tuning parameters ($c_0$ and $c_1$) representing the sum of the net longwave radiation balance and the turbulent sensible heat exchange."*

**[RC2.GC6]** The results chapter presents relevant results and also contains elements of a discussion. Please rename the chapter results and discussion or even better strictly separate results and discussion. The discussion should be a bit more complete. Please also revise whether 4.1. really belongs to the results. It might also be moved to the method chapter, where a section about calibration could be added.

We agree with the reviewer and we changed the title of section 4 into "*Results and discussion*". Further, as suggested, we moved the information of section 4.1 to the method section (see RC2.GC4).

**[RC2.GC7]** The discussion in chapter 5 could be extended (see also my comments above on the boundary conditions). Furthermore, the uncertainties of the study could be contextualised by referring to previous studies. In section 5.2., I think it would be interesting to discuss a potential firn warning on Grigoriev (as a response to different parameter disturbances) than just saying that the site remains cold. The statement from the abstract 'a detailed analysis concerning the influence of temperature and precipitation changes at the surface reveals that the thermal structure of both ice bodies is not a constant over time, with recent climate change causing more temperate ice in higher areas' could be underlined more with the discussion in 5.2.

Done. We added a discussion about the uncertainties of $P_{max}$ and $s_{max}$ (see RC2.66) and we added additional references to previous studies on the parametrisations we used (e.g. Wright et al., 2007). Besides, we mentioned the warming of the surface layer in the highest areas (see RC2.67).

**Specific comments:**

**[RC2.1]** L8/9 delete the first sentence of the abstract and complete the second: "An accurate knowledge of the thermal regime of glaciers and ice caps is important to understand their dynamic and response to climate change, and to model their evolution."

Done.

**[RC2.2]** L17-19 statement needs to be corrected/rephrased. Not sure, whether the term superimposed ice is correct here. I expect latent heat release being relevant also from other processes than the formation of superimposed ice. Refreezing of melt water above and also below the last summer horizon (internal accumulation) are probably very important processes as described for Sary Tor in Dyurgerov & Mikhalenko (1995) and also described in Kronenberg et al. (2016), and indicated by ice lenses and layers in Grigoriev cores (e.g. Takeuchi et al., 2014; Thompson et al., 1997).

We agree with the reviewer and changed the text from "superimposed ice" to "*refreezing meltwater*". Next to that, we added on lines 659-662:

*"Refreezing of meltwater below the last summer horizon can occur if not all pore spaces of the existing snow cover are occupied (< 60%), as it was shown to be an important process for glaciers and ice caps in the area (Dyurgerov and Mikhalenko, 1995; Takeuchi et al., 2014; Kronenberg et al., 2016)."*

**[RC2.3]** L19 The use of the term 'ice surface temperature' might be confusing, see also later comment.

See comment RC2.26. We changed the text into "*surface layer temperature*" instead of ice surface temperature.

**[RC2.4]** L30-32 Please be a bit more specific here. I suggest to add one or two sentences + references why such knowledge is needed and how the thermal regime affects the response of ice bodies to climate change.

See RC2.11. We replaced several sentences describing the importance of the thermal regime to the first paragraph of the introduction. Furthermore, we added an additional reference to *Hambrey and Glasser, 2012.*

**[RC2.5]** L38&L41 As Abramov glacier is located in the Pamir Alay and not in the Tien Shan, I am not sure whether it should be mentioned here. If so, I suggest to write something like 'Abramov glacier located in the nearby Pamir Alay and rather refer to the subsurface temperature measurements directly (Kislov et al., 1977), which indicated cold temperatures in the ablation area and temperate conditions in the accumulation area. Please also note, that in the uppermost meters, seasonal cooling is quite substantial on Abramov glacier, but all the cold content in the accumulation area is 'consumed' by refreezing of melt water leading to temperate conditions (cf. Kronenberg et al., 2022).

We thank the reviewer for this suggestion. We reordered this sentence and we added: "*and on the Abramov glacier, located in the nearby Pamir Alay (Kislov et al., 1977)*"

**[RC2.6]** L51-54 Please verify statements and cited references here. At least Kronenberg et al. (2021) does not state this. In the infiltration-congelation zone, the refreezing of melt water is limited by the amount of available pore space and not by temperatures which are rather cold (than temperate) here (Shumskii, 1964).

Done. We removed "heat up the ice temperature to zero degree" and removed a reference to Kronenberg et al. (2021).

**[RC2.7]** L54-56 statement unclear. Do you refer to the infiltration zone as defined in Kotlyakov, (1984); Krenke (1982)? What is thawed melt-water? Depending on the site, infiltration-congelation zone is the lower-most zone (cf. Shumskii, 1964).

We refer to the infiltration zone as described in Maohuan (1982) for the Ürümqi glacier in the Chinese Tien Shan, which is the lowest part of the accumulation zone where meltwater infiltrates in the snow layer. We removed "In the infiltration zone" and replaced this by "*In the lowest part* of the accumulation area". Furthermore, we removed "thawed" in this sentence.

**[RC2.8]** L80 calving -> dry calving

Done

**[RC2.9]** L86 delete 'on the other hand'

Done

**[RC2.10]** L86 what is small? Better provide area in km$^2$

Done. We added "*(~ 2 km$^2$)*".

**[RC2.11]** L106-114 this rather belongs to the introduction than to the data section. Also, it seems that some statements are repeated from above

We agree with the reviewer and removed this part from the data section and added most of the sentences to the first paragraph of the introduction (*see lines 31-44*).

**[RC2.12]** L114 reference needed

We added different references: *(Dikikh, 1965; Vasilenko et al., 1988; Thompson et al., 1993; Arkhipov et al., 2004; Takeuchi et al., 2014)*

**[RC2.13]** L120 warm -> temperate

Done

**[RC2.14]** L135 last -> deep

Done

**[RC2.15]** L135 on the top -> near to the summit

Done

**[RC2.16]** L142 delete last statement

Done

**[RC2.17]** L151 what kind of mountain glaciers? What climatic environment? Maybe also provide this information in the introduction…

Done. We replaced this section to the introduction and provided additional information about the glacier and ice cap on which the model has been applied (see lines 125-127).

**[RC2.18]** L151-153 was there a study using the same model and providing a detailed description? Would be good to state here, which study this was and write something like: "In the following the model is briefly described. Please refer to xx and references therein for a more detailed description. "Despite this, make sure that the reader knows the basis of your choices such as chosen parameters etc.

Done. We followed your suggestion and referred to Fürst et al. 2011. Furthermore, we added an annotation table (Table 1).

**[RC2.19]** L164 why 3? Reference

Done, we added: "*, its most common value used in previous studies (Zekollari et al., 2013)."*

**[RC2.20]** L164 and also elsewhere always write (eq. 2) and not just (2) when referring to equations

Done

**[RC2.21]** L167 I am not sure whether this should be written here or whether there might a better place to describe the spatial resolution. Might also be good to recall what the average layer thickness is (for both sites)

We decided to keep the description of the spatial resolution (horizontal and vertical) in this section. We agree to mention that for both ice masses, the average layer thickness resembles to 3-5 m i.e.

**[RC2.22]** L175 provide reference

Done. We added a reference to *Fürst et al. (2011)*
* * *
**[RC2.23]** L202 provide typical values of m for typical conditions…

This depends on how the A($T_{pmp}$) equation is formulated. We added a reference to Huybrechts (1991) and Zekollari et al. (2013). The value of m has to be evaluated experimentally, but it is generally comprised between 1 and 10. We added in line 337:

"*and which is generally comprised between 1 and 10 (Huybrechts et al., 1991; Zekollari et al., 2013)*"
* * *
**[RC2.24]** L235 (Barandun et al., 2015) is in the Pamir Alay. I don't know whether this is relevant here.

Agree. We removed the reference to Barandun et al. (2015). Furthermore, we added two additional references to studies in which more details are given about measurements of the geothermal heat flux in the Kyrgyz Tien Shan: *"Duchkov et al., 2001; Vermeesh et al., 2004"*
* * *
**[RC2.25]** L249-251 verify statement/definition of the zone (see also previous comments)

We replaced "recongelation" by "congelation" and we also referred to this zone as the infiltration recrystallisation following Shumskiy (1955). To clarify this, we also added "*and where the meltwater refreezes again.*"
* * *
**[RC2.26]** L256 Here and probably also later, it might be clarifying to write 'surface layer' instead of 'surface' e.g. "…meltwater refreezing and its warming effect on the glacier surface layer"

We thank the reviewer for this useful suggestion, and we replaced "*surface*" with "*surface layer*".
* * *
**[RC2.28]** L266 Rather provide an example from a glacier or better introduce this, otherwise the reader is very surprised that you start to talk about permafrost here..)

We agree with the reviewer. To avoid confusion for readers, we chose to delete this sentence and end the paragraph with a reference to Hooke et al. 1983.
* * *
**[RC2.29]** L276-280 Make sure, that this paragraph really states what you would like to, what is not fully clear as it is written now).

We rephrased this paragraph in order to make it more clear for the reader. It now reads"

"A last effect that can influence the surface layer temperature is the percolation of water through crevasses and moulins. This water can provide heat of friction, or it can refreeze, contributing to ice temperature warming (Philips et al., 2010; Gilbert et al., 2020). These processes are not considered in this study because neither for the Grigoriev nor for the Sary-Tor glacier significantly large areas with crevasses or moulins were detected. Observations show that meltwater on both ice masses is mainly discharged supraglacially."

**[RC2.30]** L285 'for all the effects described above' is unclear

We replaced this sentence in line 559:

*"accounting for the effects of refreezing meltwater and snow insulation, which are described above"*
* * *
**[RC2.31]** L290 why sinusoidal and not monthly lapse rates provided by (Aizen et al., 1995)

We used a sinusoidal description of the lapse rates based on the monthly lapse rates provided by Aizen et al. 1995 to ensure a smooth transition of the lapse rates during the year. We added on line 577:

*"to ensure a smooth transition of the lapse rates during the year"*
* * *
**[RC2.32]** L293 refer to section describing mass balance model (in more detail than it is currently done).

Done. We added *"see section 3.4."*
* * *
**[RC2.33]** L297 It would be good to justify this choice, as most precipitation is falling during summer months cf.(Dyurgerov et al., 1994; Kronenberg et al., 2016)

Thank you for this comment. We changed it to "*end of spring*" as the maximum snow depth is usually reached in May/June. It is indeed true that most precipitation falls in the summer, but during summer, melt compensates for the accumulation, so that the snow thickness decreases (strongly) at the (largest part of the) surface of the ice masses between June to September.
* * *
**[RC2.34]** L342-343: Why does the water refreeze at the end of the mass balance year? What are the implications of this assumption and of the assumption of the chosen threshold of 60 %? How does this estimation compare to more sophisticated simulations of refreezing? And what is about refreezing below the last summer surface?

The water does not only refreeze at the end of the mass balance year. It can also refreeze during the year if a snowpack is present, and the amount of retained and refrozen water is smaller than 60% of the snowpack (pore space). The $r_{rem}$ value corresponds to the amount of refrozen meltwater remaining at the end of the mass balance year.

Refreezing of meltwater below the last summer surface can occur if less than 60% of the pore spaces of the snow layer are occupied. We added a sentence in the manuscript to make this clearer in lines 656-657:

*"we assume all snowmelt to percolate into the snow cover and to be retained and to refreeze until a maximum amount of $P_{max}$ (taken as 60% of the snow depth) is reached (Reeh, 1991; Huybrechts et al., 1991)."*

The chosen threshold of 60% is adopted from Reeh (1991) and it corresponds to the average amount of pore spaces in snow. If this value would be smaller or larger, than $w_f$ would also change (as it is calibrated).

**[RC2.35]** L347-348: Can this choice be underlined by in situ measurements? (See also comment above. To my knowledge, most precipitation is falling during summer months and I would therefore assume substantial snow mass being accumulated after May.)

In summer months, snow falls episodically but in general, the snow thickness does not increase (substantially). Therefore, the maximum snow thickness is typically reached in May, and sometimes in June. This is approved by local measurements at stakes and snow pits.

**[RC2.36]** L375: Can you please quantify the uncertainties of this assumption? Why not use a DEM based on optical data?

We only use the reconstructed DEM as a starting point in the model. The input DEM is therefore only used to calibrate the enhancement factor (m). Errors in the created DEM do not affect the determined m-value. It was therefore beyond the focus of this study to reconstruct an optical-based 1990 DEM.

**[RC2.37]** Section 3.5.4: I suggest to delete this section and to add the relevant information to section where you describe the outlines.

Done. We added this section to section 3.5.1 in which we describe the creation and optimisation of the outlines.

**[RC2.38]** L421 15% larger than what?

We added "*larger than the present-day area*" in the text.

**[RC2.39]** L422 ~30% more than what?

We added "*larger than the present-day volume*" in the text.

**[RC2.40]** L423 The values given for Sary Tor here are different from the ones given above. Is this due to your correction based on melt? I don't fully understand this correction and it is difficult to get, why the values are presented once more.

The values given here for Sary-Tor glacier as well as for the Grigoriev ice cap correspond to the ice masses reconstructed for 1990.

**[RC2.41]** L426 refer to a map which shows this eastern tributary

Done. We added "*visible at the right panel of Figure 3".*

**[RC2.42]** L438-443 These are not results. Also, Grigoriev ice cap is strongly wind-exposed and wind erosion may also play a major role here.

As suggested in RC2.GC6, we have opted to call section 4 "Results and discussion" so that this information now fits here because the gradients found are considered in a broader perspective. As suggested, we added that Grigoriev is more wind exposed in line 651:

*", and making it more exposed to wind erosion."*

**[RC2.43]** L450/451 It is not very clear here, where these values are coming from. Are they optimised to obtain a better fit between the measured and modelled glacier extent for 1990. (I assume so from the statement in L397). This somehow implies that the mass balance model does not provide suitable mass balance estimates. Does this not also question the mass balance model estimates of the refreezing?

This mass balance correction implies that the reconstructed ice masses for 1990 are not totally in equilibrium with the 1960-1990 climatic conditions. Therefore, we adjusted the average SMB slightly to have an equilibrium state matching the 1990 outline (as derived from satellite images). Therefore, it does not imply that the mass balance is incorrect but that the average 1960-1990 climatic conditions would result in a slightly different geometry.

We added in lines 862-863:

*"This mass balance correction implies that the reconstructed ice masses for 1990 are not totally in equilibrium with the 1960-1990 climatic conditions."*

**[RC2.44]** L466 please provide units. And maybe recall, what wf stands for (firn warming parameter).

We added "*firn warming parameter*" and also added units.

**[RC2.45]** L466ff this is rather a discussion than results.

See RC2.GC6, we renamed this section Results and discussion.

**[RC2.46]** L467 I doubt that this is a valid argument here. Refreezing won't take part at depth, but only in the firn. Furthermore, refreezing may be reduced due impermeable ice layers cf. (Machguth et al., 2016)

We agree with the reviewer and removed this statement.

**[RC2.47]** L472 please provide units.

Done

**[RC2.48]** L482 I think this is rather related to the lack of firn pore space than due to a thin snow pack. The firn is likely thinner and with higher densities in the lower accumulation area.

Agree. We removed this sentence.

**[RC2.49]** L486 Summer precipitation is also mostly snow (cf. e.g. Dyurgerov et al., 1994, Kronenberg 2016).

We can confirm this by our field observations. Summer precipitation (showers) mostly consist of hail and snow (especially higher up the glaciers), but they do not increase the snowpack substantially. In the ablation area, snow already melts after a couple of hours/days. Therefore, we only consider the insolation effect of snow in winter/spring and use the maximum snow depth between 1 October and 30 May.

**[RC2.50]** L488 surface -> surface conditions (2x)

Adjusted

**[RC2.51]** L493 ensures a larger cooling of the surface ◊ allows for a larger surface cooling

Modified

**[RC2.52]** L512 'The enhancement factor is set at 3 for the Grigoriev ice cap and at 4 for the Sary-Tor glacier, which appears to be the optimal value to match with the reconstructed ice thickness of 1990.' Is not really a result. I suggest to rather provide this information within the method chapter adding a section where you describe the calibration.

Done. We added a section (3.6) about the calibration.

**[RC2.53]** L515-522 Please shorten this paragraph saying the modelled and reconstructed topographies match well and discrepancies are mainly occurring in areas where no GPR measurements exist.

We shortened this paragraph substantially by following your suggestion.

**[RC2.54]** L525 How do your measurements of the ice thickness compare to the deep core which was drilled relatively near to the summit (Takeuchi et al., 2014)

The difference between our measured ice thickness close to the summit and the deep ice core is less than 10 metres which is within the error bounds of our measurements.

**[RC2.55]** L527 Please provide a refence for the statement about the complexity of modelling ice thicknesses at ice divides

Done. We added a reference to "*Pattyn, 2003*".

**[RC2.56]** L529 Please provide a reference for the enhancement factor of 1 corresponding to -5°C (rather write °C than 'degrees').

We changed degrees in °C and we added a reference to *Huybrechts and Oerlemans (1988)*.
* * *
**[RC2.57]** L542 Please refer to the map in fig 8c it here.

Done. We added "c".
* * *
**[RC2.58]** L561 Please add here, when these observations were performed.

Done. We added that the observations were performed in 2018.
* * *
**[RC2.59]** L572 Please use full variable names and give short forms in brackets.

Done.
* * *
**[RC2.59]** L573: Should an improved transport of geothermal heat not rather lead to a warming?

No. By this statement we mean that thinner ice transports geothermal heat more easily, resulting in a lower ice temperature gradient (and hence resulting in cooler temperatures at depth). We removed "flux" to avoid ambiguity.
* * *
**[RC2.60]** L576 ensures -> shows?

Modified.
* * *
**[RC2.61]** L579-580 Please rephrase 'This ensures a greater stiffness which causes the ice intrinsically slower to deform.'

Done. We replaced this sentence by "*This causes the ice to deform more slowly*"
* * *
**[RC2.62]** L584/585 Not clear, why temperate ice causes the infiltration of water. Water can either infiltrate into ice through crevasses, moulins or infiltrate into the firn unless an impermeable ice layer is reached within the firn (Machguth et al., 2016). Such infiltration occurs in cold and temperate firn (Shumskii, 1964).

The reviewer is right that water can mainly infiltrate into ice through crevasses, moulins or infiltrate into the snow and firn. However, temperate ice contains networks of intragranular veins where the flow of water is possible if it is not blocked by air bubbles (Nye and Frank, 1973; Lliboutry et al., 1996). For cold ice in contrast, veins and cracks usually refreeze within hours making it fracture-free and impermeable. This facilitates the formation of deeply incised and persistent melt water streams and supraglacial lakes. The latter are often an indication of cold ice. Hence, we added in the text more information on lines 1131-1135:

*"as temperate ice contains networks of intragranular veins where the flow of water is possible if it is not blocked by air bubbles (Nye and Frank, 1973; Lliboutry, 1996; Ryser et al., 2013). Cold ice in contrast, is fracture free (water refreezes within hours in smaller cracks), making the ice impermeable to water which facilitates the formation of persistent and deeply incised water streams and supraglacial lakes (Boon and Sharp, 2003; Ryser et al., 2013)."*
* * *
**[RC2.63]** L587 'This is the case for the entire surface of the Grigoriev ice cap'. This is not correct. Water percolation of into the subsurface also occurs on Grigoriev as e.g. visible from ice lenses within the firn in different cores from (Takeuchi et al., 2014; Thompson et al., 1997)

Thank you for this remark. We modified this in the text on lines 1136-1138:

*"This is the case for the largest part of the surface of the Grigoriev ice cap. Only in the upper part of the ice cap, ice lenses were observed when boreholes were made (Thompson et al., 1997; Takeuchi et al., 2014)."*
* * *
**[RC2.64]** L613/614 This statement needs a reference. And, also, it is not fully clear why this is written here. I suggest to rephrase the paragraph so that the argumentation becomes clearer.

Done.
* * *
**[RC2.65]** L630 Could the mismatch in fig. 11 between simulations and measurements not also be (partly) related to a wrong lower boundary condition and/or the used temperature gradient? Would be interesting to visualize of the corresponding sensitivity experiment here.

See RC2.GC2. We applied a topographical correction for the geothermal heat flux. Besides, we added two more temperature profiles created with 50% and 200% of the average geothermal heat flux. We added in the caption in lines 1160-1164:

*"The purple profile is a time-dependent temperature profile obtained by using 50% of the average geothermal while the green dashed line corresponds to the time-dependent temperature profile obtained by using 200% of the average geothermal heat flux."*

**[RC2.66]** L675 the way, how wf and especially is are calculated also has uncertainties. This should be discussed here, as the overall temperature regime seems to be very sensitive to is

We included several sentences and refer to the study of Wright et al. (2007) where different methods for the $P_{max}$ value are described, in lines 1261-1271

*"Concerning the refreezing of meltwater, which appeared to be crucial for the determination of the surface layer temperatures, it must be emphasised that in this study, a simple parametrisation is used with a constant $P_{max}$ value (Reeh, 1991; Wright et al., 2007), and to model its warming effect on the surface layer (Huybrechts et al., 1991; Zekollari et al., 2017). Other, more complex, approaches and models exist (Wright et al., 2007; Reijmer et al., 2012). Nevertheless, in this study, direct and multiple temperature observations in boreholes were used to calibrate the parameters of the applied parameterisation, which means that the effects of the uncertainties of the parametrisation on the results are expected to be limited. With regard to $s_{max}$ and the snow insulation method, the largest uncertainty arises from the date of the maximum snow depth. In some years, the maximum snow depth is only reached after the end of May, for example, when substantial snow accumulates during the beginning of the summer season, or when the ablation season initiates later. In addition, significant accumulation of snow in autumn/winter is sometimes delayed, causing the snow to have a smaller insulating effect in this period (Hooke, 1983)."*

**[RC2.67]** L705 From Figure 12 it seems, that warming temperatures cause an increase of the percentage of temperate ice on Grigoriev. So, the statement that it remains cold should be corrected. Also, do your simulation show a warming happening with warmer temperatures. Even if temperatures stay below the melting point, a warming may be happening – would be interesting to briefly discuss this here and maybe put into context with observed and modelled fin warning in the alps (Hoelzle et al., 2011; Mattea et al., 2021; Vincent et al., 2020)

Figure 12 shows that for increasing temperatures, when the precipitation is kept constant, the percentage of temperate ice remains negligible. Only for precipitation increase, while keeping temperatures fixed (unlikely), the fraction of temperate ice becomes larger. This is the reason why we state that the Grigoriev ice cap remains cold.

Thank you for this suggestion. Our simulations indeed indicate a warming at higher altitudes. We rephrased this paragraph and added on lines 1404-1406:

*"… the enhanced formation of refrozen water (and the associated release of latent heat) (see section 4.4.2). Such a firn warming has also been observed and modelled at other glaciers (Hoelzle et al., 2011; Vincent et al., 2020; Mattea et al., 2021)."*

**[RC2.68]** L717 not only the formation of superimposed ice (which happens at the edge of the accumulation area) releases heat, but also refreezing of melt water within the firn is a relevant warming process.

Clear. We changed "superimposed ice" by "*refrozen water*". See RC2.67

**[RC2.69]** L718 Please change 'This has already been observed at the Abramov glacier in the southwestern Tien Shan (Kronenberg et al., 2020).to 'Evidence of a precipitation increase has been observed on Abramov glacier in the Pamir Alay (Kronenberg et al., 2021)'

Done

**[RC2.70]** L735 'reduced formation of refrozen meltwater at the end of summer' -> 'reduced refreezing of melt water' (Refreezing is an ongoing process, not restricted to the end of summer. Especially on temperate sites, it happens at the beginning of the melt season (cf. Kronenberg et al., 2022)

Done

**[RC2.71]** L737 This may be the case in the model applied here, which does not account for firn. But is not true in general. Melt water is known to infiltrate to the subsurface also under cold conditions (Shumskii, 1964). Please clarify here.

See RC2.62. We added "*through intragranular veins*".

**[RC2.72]** L766 Sary-Tor -> Grigoriev

This is correct. Victor Popovnin provided the mass balance data of the Sary-Tor glacier.

**[RC2.73]** L870 2020 -> 2021

Done.

**Technical corrections**

**[RC2.74]** References: Please add references to statements where they are missing and make sure that all references are given following the journal's guidelines. Sometimes, the year is provided in brackets, sometimes not.

Done.

**[RC2.75]** The language of manuscript could be improved I recommend a proofreading by a native speaker.

We polished the language, and we believe it improved the quality of the manuscript.

**General comment on the tables:**

**[RC2.76]** Please homogenize the table layout. I suggest to use a typical layout without coloured lines, shaded areas etc.

Done.

**General comment on figures:**

**[RC2.77]** Please add panel letters to all the figures which show more than one plot/map and refer to those letters in the captions. Use larger fonts in legends and labels. Plot legends aside and not on the top of maps/plots and show them in a reasonable size (they are small and difficult to read). Place labels on maps more carefully so that they are better visible. Complete information in caption (year of glacier outlines is usually missing, say that the legend is for all the subplots etc.).

Done.

**[RC2.78]** Table 2: provide units also in the table (and not only in the caption). Does 1986-1987 refer to the mass balance year starting from 1.10.1986 and ending on the 30.9.1987? If so rather write 1986/87. And replace 'Average 1963-1989' with 'average 1963/64-1988/89'. Provide reference in a sperate column. Also, it seems strange that the glacier wide value is given in the centre. Either provide it first or last. And 'at the front' is very unspecific. Due you refer to a point measurement here or to some averaged value? Please show on a map where this is and specify, whether always the location is used.

Done. And we added "Average $m_s$ < 4200 m a.s.l.) because it represents an average surface mass balance.

**[RC2.79]** Figure 1: add legend with information about of background topography

Done.

**[RC2.80]** Figure 3: Zoom in to Sary Tor. The glacier is quite small and you loose space by showing its surroundings. Visualize core locations on Grigoriev map.

Thank you for the suggestion. We zoomed in to the Sary-Tor glacier and we added the core location of the 2007 borehole. As most of the other core locations were near/on the summit, we decided to not include these on the figure for clarity.

**[RC2.81]** Figure 5b display x and y axes with the same length.

Done.

**[RC2.82]** Figure 6: Provide x/y axes for all panels or say in the captions that they stand for all. Improve legend!

Thank you for the suggestion. We added this sentence to the caption and replaced the legends:

*"The x and y-axes on the left panels stand for the other panels as well."*

**[RC2.83]** Figure 8: If D in panel d corresponds to D in panel c, 'A' should be replaced with 'C' in panel c.

Done.

**[RC2.84]** Figure 9: cross-reference in caption wrong (should be figure 8 or 7). Add letters: A,B,C,D

The cross-reference is correct. However, for clarity, we modified *"indicated in" to "used in"*. We added letters A,B,C,D.

**[RC2.85]** Figure 10 Clarify the colour bar. Is the darkest red colour tone in the left panel referring to temperatures at 0°C. What does the second class stand for? Values between -0.5 and -1°C?
Plot the simulated boundary between temperate and cold ice in the right panel. And show both panels at equal spatial resolution

The darkest red colour refers indeed to temperatures at and very close to 0°C. The second class is for values between -0.01°C and -1°C. We added this in the legend. We also added the simulated boundary between temperate and cold ice in the left and right panel and modified the figures to have equal spatial resolution.

[Figure]

**[RC2.86]** Figure 11: was the purple profile simulated by a stepwise temperature increase or was the change applied once?

The purple profile is a steady state temperature profile, simulated by imposing the average 1980-2010 climatic conditions.

**[RC2.87]** Figure 12: homogenize font sizes of axes labels

Done.